# Fast kernel methods: Sobolev, physics-informed, and additive models

## Abstract

Kernel methods are powerful tools in statistical learning, but their cubic complexity in the sample size $n$ limits their use on large-scale datasets. In this work, we introduce a scalable framework for kernel regression with $\mathcal{O}(n \log n)$ complexity, fully leveraging GPU acceleration. The approach is based on a Fourier representation of kernels combined with non-uniform fast Fourier transforms (NUFFT), enabling exact, fast, and memory-efficient computations. We instantiate our framework in three settings: Sobolev kernel regression, physics-informed regression, and additive models. The proposed estimators are shown to achieve minimax convergence rates, consistent with classical kernel theory. Empirical results demonstrate that our methods can process up to tens of billions of samples within minutes, providing both statistical accuracy and computational scalability. These contributions establish a flexible approach, paving the way for the routine application of kernel methods in large-scale learning tasks.

## 1 Introduction

**Kernel methods.** Kernel methods play a central role in statistical learning and nonparametric regression, providing flexible and theoretically sound tools for modeling complex data structures (Steinwart & Christmann, 2008). Despite their appeal, their practical application is often hindered by severe computational constraints: standard kernel ridge regression has a time complexity of $\mathcal{O}(n^3)$ and a memory cost of $\mathcal{O}(n^2)$, making it unsuitable for large-scale datasets (Bach, 2024, Chapter 7). Several approximation schemes, such as Nyström methods and random feature expansions, have been proposed to reduce this burden, reducing the computational cost to $\mathcal{O}(n^{3/2})$. However, these approximations introduce additional error terms, complicating theoretical guarantees and often degrading empirical performance (Meanti et al., 2020).

**Contributions.** We show that a broad class of kernel estimators can be trained exactly at a substantially reduced computational cost, achieving $\mathcal{O}(n \log n)$ complexity in terms of both time and memory. This improvement is made possible by taking advantage of the structural properties of kernels that can be expanded on a Fourier basis. Furthermore, our approach lends itself well to parallel computation and can be efficiently implemented on modern GPU architectures. These advances are a significant step forward in both the theory and practice of kernel methods, with clear potential for routine application to large-scale datasets. A key component of our approach is the recent implementation of the non-uniform fast Fourier transform on GPU (NUFFT, Shih et al., 2021). The resulting computational efficiency enables the processing of tens of billions of data points in under a minute on a standard GPU, whereas conventional kernel approaches typically fail to scale beyond a few hundred thousand samples (Meanti et al., 2020; Doumèche et al., 2025a).

We illustrate our framework through three fundamental kernel learning tasks: Sobolev regression (Nemirovski, 2000), physics-informed regression (Doumèche et al., 2025b), and additive model regression (Hastie & Tibshirani, 1986). The ability to process massive datasets allows us to systematically assess the performance of the estimators, clearly highlighting the benefits of our approach. In particular, we introduce a kernel- and Fourier-based additive model with a GPU implementation that combines statistical accuracy, favorable scaling in the input dimension $d$, and high computational efficiency into a unified and theoretically sound framework. Table 1 completes Table 1 from (Rudi

et al., 2017) and compares the performance of our algorithms to state-of-the-art kernel speed-up methods.

| Algorithm | train time | kernel evals | memory | test time |
|---|---|---|---|---|
| SVM / KRR + direct method | $n^3$ | $n^2$ | $n^2$ | $n$ |
| KRR + iterative (Caponnetto & Yao, 2010) | $n^2\sqrt{n}$ | $n$ | $n^2$ | $n^2$ |
| Doubly stochastic (Dai et al., 2014) | $n^2\sqrt{n}$ | $n^2\sqrt{n}$ | $n$ | $n$ |
| Pegasos / KRR + SGD (Shalev-Shwartz et al., 2011) | $n^2$ | $n^2$ | $n$ | $n$ |
| KRR + iterative + preconditioning (Ma & Belkin, 2017) | $n^2$ | $n^2$ | $n$ | $n$ |
| Divide & Conquer (Zhang et al., 2013) | $n^2$ | $\sqrt{n}$ | $n$ | $n$ |
| Nyström / Random Features (Williams & Seeger, 2000) | $n^2$ | $\sqrt{n}$ | $n$ | $\sqrt{n}$ |
| Nyström + iterative (Camoriano et al., 2016) | $n^2$ | $\sqrt{n}$ | $n$ | $\sqrt{n}$ |
| Nyström + SGD (Dieuleveut & Bach, 2014) | $n^2$ | $\sqrt{n}$ | $n$ | $\sqrt{n}$ |
| FALKON (Rudi et al., 2017) | $n\sqrt{n}$ | $n\sqrt{n}$ | $n$ | $\sqrt{n}$ |
| **NUFFT kernels** (this article) | $n$ | $N.A.$ | $n$ | $n^{1/(2s+d)}$ |

Table 1: Comparison of computational complexity of kernel methods. Logarithmic terms are not showed. In this paper, $s > d/2$, so that $n^{1/(2s+d)} = o(n^{1/2})$ as soon as $s \geq 1$.

**Related works.**    The usage of NUFFT to reduce the kernel training cost to $O(n \log n)$ has been pioneered by Barnett et al. (2024); Greengard (2025); Greengard et al. (2025) for translation-invariant kernels, i.e., kernels such that $K(x, y) = k(x - y)$ for some function $k$, and such that the Fourier series of $k$ is known. Neither of these two conditions is satisfied by physics-informed kernels, for which the geometry of the problem is extremely important (for a counter-example, see Proposition 5.1 of Doumèche et al. (2024)). Instead of focusing on the Fourier representation of the kernel, the method developed in the article can be applied whenever the kernel norm can be quickly computed.

Most of the kernel regression literature focuses on the asymptotic properties of infinite dimensional RKHS, such as Caponnetto & Vito (2007). This approach is mainly motivated by kernel universality, which is often considered as a requirement for a kernel function. Our point of view here is different and focuses on finding the best estimator for a given regression task. In particular, we show that the $L^2$ Fourier kernel of Proposition 3.2, which does not approximate any infinite-dimensional kernel, provably achieves better performance than the Sobolev kernel of Proposition 3.1 when learning a Sobolev function.

Up to our knowledge, our contribution is the first one to provide an $O(n \log n)$ algorithm for additive regression. We believe it to be a key methodological contribution, given that the current state-of-the-art method runs in $O(n^{7/5})$, and considering how massively used additive models are in statistics (Wood et al., 2014).

**Layout.**    The paper is organized as follows. In Section 2, we introduce the Fourier-based kernel approximation and the fast optimization strategy. Section 3 develops the Sobolev kernel regression framework and establishes its statistical guarantees. Section 4 extends the method to physics-informed regression, while Section 5 presents the additive model formulation.

## 2    COMMON SETTING

### 2.1    NONPARAMETRIC REGRESSION WITH KERNEL METHODS

**General assumptions.**    Throughout this article, we adopt the standard nonparametric regression setting, in which the objective is to estimate an unknown function $f^\star$ from $n$ i.i.d. observations $(X_1, Y_1), \ldots, (X_n, Y_n)$ of a random pair $(X, Y)$, under the following assumptions:

(i) The input variable $X$ takes values in an open domain $\Omega \subseteq \mathbb{R}^d$ for some $d \geq 1$, and the output $Y$ is real-valued;

(ii) The output is modeled as $Y = f^\star(X) + \varepsilon$, where the noise term $\varepsilon$ satisfies $\mathbb{E}(\varepsilon \mid X) = 0$ and $\mathbb{E}(\varepsilon^2 \mid X) \leq \sigma^2$ for some $\sigma > 0$.

To guarantee that $f^\star$ is learnable from data, we impose the following regularity assumptions:

(iii) The target function belongs to a Sobolev space of smoothness $s > d/2$, that is, $f^\star \in H^s(\Omega)$ for some known $s$;

(iv) The domain $\Omega$ is a bounded Lipschitz domain of $\mathbb{R}^d$ (see, e.g., Agranovich, 2015), included in the hypercube $[-L, L]^d$ for some $L > 0$.

Recall that $H^s(\Omega)$ denotes the Sobolev space of functions whose weak derivatives up to order $s$ belong to $L^2(\Omega)$. This space includes, for example, the Hölder space $C^s(\bar{\Omega})$ of continuously differentiable functions with bounded derivatives. The Lipschitz assumption allows the boundary of the domain to be non-differentiable, encompassing a broad class of bounded sets, including smooth manifolds and common geometries, such as hypercubes.

**Kernel methods.** In kernel-based methods, this setting naturally leads to the minimization of the regularized empirical risk functional

$$\mathcal{R}(f) = \frac{1}{n} \sum_{j=1}^n (f(X_j) - Y_j)^2 + \lambda \|f\|_{\mathcal{H}}^2,$$

over a reproducing kernel Hilbert space (RKHS) $\mathcal{H} \subseteq H^s(\Omega)$, where $\lambda > 0$ is a regularization parameter. The norm $\|\cdot\|_{\mathcal{H}}$ serves as a regularizer and reflects the prior assumption that the true function $f^\star$ has a bounded RKHS norm, i.e., $\|f^\star\|_{\mathcal{H}} < \infty$. The resulting estimator $\hat{f} = \arg\min_{f \in \mathcal{H}} \mathcal{R}(f)$ serves as an estimator of the target function $f^\star$. A classical result in kernel theory states that the estimator $\hat{f}$ can be computed in closed form as $\hat{f}(x) = \big(K(X_1, x), \dots, K(X_n, x)\big)(\mathbb{K}_n + \lambda I)^{-1}\mathbb{Y}$, where the function $K$ is the reproducing kernel associated with the RKHS $\mathcal{H}$, and $\mathbb{K}_n$ is the $n \times n$ kernel matrix with entries $(\mathbb{K}_n)_{j_1,j_2} = K(X_{j_1}, X_{j_2})$ (Bach, 2024, Chapter 7). The main computational bottleneck of this approach lies in the analytical computation of the kernel function $K$ and in the inversion of the kernel matrix $\mathbb{K}_n$, which incurs a cost of $\mathcal{O}(n^3)$ in time and $\mathcal{O}(n^2)$ in memory.

**Finite-dimensional approximation.** To make the computation of the kernel estimator more tractable, we consider a finite-dimensional approximation of $\mathcal{H}$ through a finite basis $\phi_1, \dots, \phi_D$ with $D \in \mathbb{N}^\star$ elements. Thus, any function $f \in \mathcal{H}$ can be approximated by a function $f_\theta$ of the form

$$f_\theta(x) = \theta_1 \bar{\phi}_1(x) + \cdots + \theta_D \bar{\phi}_D(x) = \langle \phi(x), \theta \rangle, \tag{1}$$

where $\phi = (\phi_1, \dots, \phi_D)^\top$ denotes the truncated feature map, $\theta \in \mathbb{C}^D$ is the parameter vector, and the inner product is defined as $\langle x, y \rangle = \sum_{k=1}^D \bar{x}_k y_k$. In the present paper, we let $\phi$ be the truncated Fourier basis composed of trigonometric functions.

Since the RKHS approximation if finite-dimensional, there is a positive-definite matrix $M$ such that, for all function $f_\theta \in \mathcal{H}$ given by Equation (1), we have $\|f_\theta\|_{\mathcal{H}}^2 = \|M\theta\|_2^2$. The empirical risk of $f_\theta$ then translates into an empirical risk $R$ on the parameter $\theta$:

$$\mathcal{R}(f_\theta) := R(\theta) = n^{-1} \|\Phi\theta - \mathbb{Y}\|_2^2 + \lambda \|M\theta\|_2^2,$$

where $\Phi = (\phi(X_1) \mid \cdots \mid \phi(X_n))^*$ is the design matrix (with complex conjugate transpose), and $\mathbb{Y} = (Y_1, \dots, Y_n)^\top \in \mathbb{R}^n$ is the response vector. Minimizing this objective yields the parameter

$$\hat{\theta} = \arg\min_{\theta \in \mathbb{C}^D} R(\theta) = (n^{-1}\Phi^*\Phi + \lambda M^*M)^{-1} n^{-1}\Phi^*\mathbb{Y}, \tag{2}$$

and the resulting estimator takes the form $\hat{f}_n(x) = \langle \phi(x), \hat{\theta} \rangle$.

## 2.2 FOURIER EXPANSION

Using a Fourier basis makes RKHS approximation errors precisely quantifiable, as shown below.

**Fourier expansion on the Sobolev space $H^s(\Omega)$.** The smoothness of a function in $H^s(\Omega)$ (endowed with any of the equivalent Sobolev norms of order $s$) is reflected in the decay rate of its Fourier coefficients (see Proposition A.1 in the appendices). Since the Fourier coefficient vector $\theta(f) \in \mathbb{C}^{\mathbb{Z}^d}$ is infinite-dimensional, it cannot be stored or manipulated directly. Thus, in line with the finite-dimensional approximation principle presented above, we truncate the Fourier expansion to a finite number of frequencies, indexed by a cutoff parameter $m \in \mathbb{N}^\star$. In other words, we consider the finite-dimensional approximation space

$$H_m = \Big\{ f : x \mapsto \sum_{k_1=-m}^{m} \cdots \sum_{k_d=-m}^{m} \theta_{k_1,\ldots,k_d} \exp\big(i\frac{\pi}{2L}\langle k, x\rangle\big),\ \theta \in \mathbb{C}^{(2m+1)^d} \Big\},$$

where $k = (k_1, \ldots, k_d) \in \mathbb{Z}^d$ is a multi-index. The dimension of this kernel Hilbert space is thus $D = (2m+1)^d$, and the truncated feature map reads

$$\phi(x) = \Big( \exp\big(-i\frac{\pi}{2L}\langle k, x\rangle\big) \Big)_{\|k\|_\infty \leq m}.$$

For any function $f \in H^s(\Omega)$, its truncated approximation $f_m \in H_m$ takes the form

$$f_m(x) = \langle \phi(x), (\theta(f)_k)_{\|k\|_\infty \leq m}\rangle.$$

**Fourier truncation.** The truncation of high-frequency Fourier modes is justified by the following approximation result. Let $\mathbb{P}_X$ denote the marginal distribution of the input variable $X$ on the domain $\Omega$. From this point onward, we assume the following regularity condition on $\mathbb{P}_X$:

(v) $\mathbb{P}_X$ is a probability measure on $\Omega$ that admits a density with respect to the Lebesgue measure, bounded above by a constant $\kappa > 0$, i.e., $\frac{d\mathbb{P}_X}{dx} \leq \kappa$.

**Proposition 2.1** (Approximation of the Fourier expansion). *Under assumption (v), there exists a constant $C > 0$, depending only on $d$ and $\Omega$, such that, for all $n \in \mathbb{N}^\star$, all $f \in H^s(\Omega)$, and all truncation levels $m \in \mathbb{N}^\star$, the following inequality holds:*

$$\|f - f_m\|_{L^2(\mathbb{P}_X)}^2 \leq C\kappa \|f\|_{H^s(\Omega)}^2 \, m^{-2s}.$$

Under Assumptions (i) to (v), any estimator $\hat{f}$ of $f^\star$ has an error $\mathbb{E}(\|f^\star - \hat{f}\|_{L^2(\mathbb{P}_X)}^2)$ at least of the order of $n^{-2s/(2s+d)}$ (e.g., Tsybakov, 2009, Theorem 2.1). This convergence rate of $2s/(2s+d)$ is known as the Sobolev minimax rate. Letting $m = n^{1/(2s+d)}$ in Proposition 2.1 is thus sufficient to achieve a precision of $n^{-2s/(2s+d)}$. Therefore, since adding extra Fourier modes would not result in an improved asymptotical precision on $\hat{f}_n$, in what follows, we consider $m = n^{1/(2s+d)}$.

## 2.3 FAST OPTIMIZATION

**From naive implementation to fast transforms.** Since the total number of Fourier modes is $(2m+1)^d = \mathcal{O}(n^{d/(2s+d)})$, a naive implementation of (2) incurs a complexity of $\mathcal{O}(n^{1+2d/(2s+d)})$, mainly due to the matrix product $\Phi^*\Phi$. While already faster than the standard $\mathcal{O}(n^3)$ kernel approach, we show how this can be further reduced to $\mathcal{O}(n\log n)$—near-optimal, as even reading the data takes $\mathcal{O}(n)$. This speedup is enabled by using complex exponential bases to discretize the kernel, which are well-suited to fast Fourier transform (FFT) algorithms. However, since the exponentials are evaluated at non-uniform points, standard FFTs are not applicable. Instead, we use the non-uniform fast Fourier transform (NUFFT), which extends FFT efficiency to irregular sampling. These algorithms can be efficiently parallelized on modern GPUs, leveraging recent advances in hardware-optimized fast summation. In what follows, we rely on the cuFINUFFT library for the GPU-based implementation of the NUFFT, restricted to dimensions 1, 2, and 3 (Shih et al., 2021).

**Covariance vector and NUFFT.** A naive computation of the covariance vector $v = \Phi^*\mathbb{Y}/n$ via explicit matrix–vector multiplication leads to a complexity of $\mathcal{O}(n^{1+d/(2s+d)})$. However, each component of $v$ can be written explicitly as

$$v_k = \frac{1}{n} \sum_{j=1}^{n} Y_j \exp\big(i\pi\langle k, \frac{X_j}{2L}\rangle\big),$$

which is the definition to the type-I NUFFT of the weighted signal $\mathbb{Y}$ with respect to the non-uniform spatial locations $(X_1, \ldots, X_n)$, leading to time and memory complexity $\mathcal{O}(n\log n)$.

**Covariance matrix and NUFFT.** A naive computation of the empirical covariance matrix $\hat{\Sigma} = \Phi^*\Phi/n$ via explicit matrix–matrix multiplication incurs a computational complexity of $\mathcal{O}(n^{1+2d/(2s+d)})$. However, one can observe that $\hat{\Sigma}$ is both Hermitian and $d$-level block Toeplitz. Specifically, it satisfies $\hat{\Sigma}^* = \hat{\Sigma}$ and, for all $k_1, k_2 \in \{-m, \ldots, m\}^d$, the relation $\hat{\Sigma}_{k_1,k_2} = \hat{\Sigma}_{0,k_2-k_1}$ (by elementary properties of the exponential function). This means that the matrix $\hat{\Sigma}$ is fully determined by its first row, i.e., the $(2m+1)^d$ values $(\hat{\Sigma}_{0,k})_{\|k\|_\infty \leq m}$. Each entry can be expressed as

$$\hat{\Sigma}_{0,k} = \frac{1}{n}\sum_{j=1}^{n} \exp\left(i\pi\langle k, \frac{X_j}{2L}\rangle\right),$$

which corresponds to the type-I NUFFT of the constant signal $(1, \ldots, 1)$ evaluated at the sample points $(X_1, \ldots, X_n)$, leading to a time and memory complexity $\mathcal{O}(n \log n)$ with GPU acceleration. Furthermore, for any $x \in \mathbb{C}^{(2m+1)^d}$, the matrix–vector product $\hat{\Sigma}x$ can be performed efficiently via FFT in $\mathcal{O}(m^d \log m) = \mathcal{O}(n^{d/(2s+d)} \log n)$ time; see Golub & Loan (Theorem 4.8.2, 2013) for the one-dimensional case and the multidimensional construction of Lee (1986).

**Matrix inversion.** We assume that for any $x \in \mathbb{C}^{(2m+1)^d}$, the matrix–vector product $M^*Mx$ can be performed in $\mathcal{O}(m^d \log m) = \mathcal{O}(n^{d/(2s+d)} \log n)$. This condition is satisfied if $M$ is diagonal or, as shown above, if $M^*M$ is a covariance matrix. Then, since the matrix–vector product $\hat{\Sigma}x$ can also be performed in $\mathcal{O}(n^{d/(2s+d)} \log n)$ time via FFT, linear systems involving $\hat{\Sigma} + \lambda M^*M$ can be solved efficiently using the conjugate gradient method. The total complexity of solving such systems is reduced to $\mathcal{O}(n^{2d/(2s+d)} \log n) = o(n)$, which is sublinear in the number of data points.

## 3 SOBOLEV REGRESSION

### 3.1 SOBOLEV KERNEL REGRESSION

In this section, we study the Sobolev kernel regression, also known as Matérn kernel, where $\mathcal{H} = H^s(\Omega)$ and the RKHS norm is given by $\|f\|_{\mathcal{H}}^2 = \sum_{k \in \mathbb{Z}^d} |\theta(f)_k|^2 (1 + \|k\|_2^{2s})$. (Since all Sobolev norms are equivalent, the choice of the Sobolev norm does not impact the rate of convergence of the kernel estimator.) In the finite-dimensional approximation using $H_m$, one has $\|f_\theta\|_{\mathcal{H}}^2 = \|S\theta\|_2^2$, where the matrix $S \in \mathbb{C}^{(2m+1)^d \times (2m+1)^d}$ is diagonal with entries $S_{k_1,k_2} = \sqrt{1 + \|k\|_2^{2s}}\mathbf{1}_{k_1=k_2}$. Using tools from kernel theory (Mourtada & Rosasco, 2022; Bach, 2024), we derive the convergence rate of the estimator $\hat{\theta}$ as a function of the sample size $n$. This result is encapsulated in the following proposition.

**Proposition 3.1** (Sobolev kernel regression). *Let $\hat{\theta}$ be the Sobolev kernel estimator* (2) *with $M = S$, i.e.,*

$$\hat{\theta} = \mathrm{argmin}_{\theta \in \mathbb{C}^{(2m+1)^d}} n^{-1}\|\Phi\theta - \mathbb{Y}\|_2^2 + \lambda\|S\theta\|_2^2$$
$$= (n^{-1}\Phi^*\Phi + \lambda S^2)^{-1}n^{-1}\Phi^*\mathbb{Y}.$$

*Under Assumptions (i)-(v),*

$$\mathbb{E}(\|\hat{f}_n - f^\star\|_{L^2(\mathbb{P}_X)}^2) \leq \inf_{\theta \in \mathbb{C}^{(2m+1)^d}} \left\{\left(2 + 6\left(1 + \frac{\alpha^2}{\lambda n}\right)^2\right)\mathbb{E}(\|f^\star - f_\theta\|_{L^2(\mathbb{P}_X)}^2) + \lambda\left(1 + \frac{\alpha^2}{\lambda n}\right)^2\|S\theta\|_2^2\right\}$$

$$+ \sigma^2 n^{-1}\left(1 + \frac{\alpha^2}{\lambda n}\right)(2m+1)^d,$$

*where $\alpha = \sum_{k \in \mathbb{Z}^d} \frac{1}{1+\|k\|_2^{2s}} < \infty$. Thus, choosing the regularization parameter in the range $n^{-1} \leq \lambda \leq n^{-2s/(2s+d)}$ and setting the truncation level as $m = n^{1/(2s+d)}$ yields the convergence rate*

$$\mathbb{E}(\|f_{\hat{\theta}} - f^\star\|_{L^2(\mathbb{P}_X)}^2) = \mathcal{O}\left(n^{-2s/(2s+d)}\right).$$

This convergence rate of $2s/(2s+d)$ matches the Sobolev minimax rate, meaning that it is the fastest rate achievable under Assumptions (i) to (v). The existence of minimax Sobolev estimators is well-known in the statistical learning literature (e.g., Nemirovski, 2000, Chapter 2). However, Proposition 3.1 establishes a clear connection between kernel methods and Fourier analysis. More importantly, since $S$ is diagonal, this estimator can be computed exactly on a GPU with complexity $\mathcal{O}(n \log n)$, as shown in Section 2.3. This enables kernel methods to scale to large datasets.

**One dimensional example.** The goal of this experiment is to demonstrate the ability of our method to efficiently handle datasets containing tens of billions of samples, while experimentally recovering the theoretical convergence rate. We consider a simple one-dimensional regression setting where the input variable $X$ is uniformly distributed over the interval $\Omega = ]0, 1[$ and the output is given by $Y = \exp(X) + \varepsilon$, with a noise $\varepsilon \sim \mathcal{N}(0, 1)$ independent of $X$. We run the Sobolev kernel estimator with smoothness parameter $s = 1$, regularization $\lambda = n^{-2/3}$, and truncation level $m = n^{1/3}$. The algorithm successfully processes $n = 10^{10}$ data points in approximately one minute on a standard GPU (NVIDIA T4), using complex-128 precision. Figure 1 reports the test error $\mathbb{E}(\|\hat{f}_{\hat{\theta}} - f^{\star}\|_{L^2(\mathbb{P}_X)}^2)$, estimated over a separate test set of $10^4$ samples, and averaged over 20 resamples. The observed performance matches the Sobolev minimax rate of $n^{-2/3}$, providing empirical evidence of the method's numerical stability and scalability.

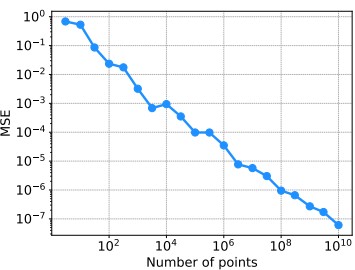

Figure 1: One-dimensional Sobolev kernel regression

### 3.2 $L^2$-FOURIER KERNEL REGRESSION

**Dependency on the smoothness $s$.** If the target function $f^{\star}$ is known to have smoothness $s$, then it also belongs to all Sobolev spaces $H^{s_2}(\Omega)$ for any $s_2 < s$. As a result, Proposition 3.1 guarantees that all Sobolev kernel estimators with smoothness parameter $s_2 < s$ are well-defined and provably converge to $f^{\star}$. From a theoretical point of view, higher values of $s$ yield faster convergence rates. However, the non-asymptotic bound in Proposition 3.1 also reveals a bias term involving $\lambda\|S\theta\|_2^2 \geq \lambda\|f^{\star}\|_{H^s(\Omega)}^2$. Overall, this term scales as $n^{-2s/(2s+d)}\|f^{\star}\|_{H^s(\Omega)}^2$, highlighting the classical trade-off inherent in the choice of $s$: larger values improve the convergence rate but come at the cost of increased dependence on the Sobolev norm of the target function.

For example, consider the function $f(x) = x^k$ on the interval $]-1, 1[$. One can compute $\|f\|_{H^1(]-1,1[)}^2 = \int_{-1}^1 f(x)^2 + f'(x)^2 \, dx = \frac{2}{2k+1} + \frac{2k^2}{2k-1}$, whereas $\|f\|_{H^k(]-1,1[)}^2 \geq \int_{-1}^1 \left(f^{(k)}(x)\right)^2 dx = 2(k!)^2$, showing that $\|f\|_{H^k(]-1,1[)}^2 \gg \|f\|_{H^1(]-1,1[)}^2$ for large $k$. This illustrates how higher-order Sobolev norms can grow rapidly and lead to high regularization bias despite improved asymptotic rates.

To mitigate the increasing regularization bias associated with large values of $s$, we propose an alternative estimator that penalizes the unweighted $\ell_2$ norm of the parameter vector, i.e., $\lambda\|\theta\|_2^2$, instead of the Sobolev-weighted norm $\lambda\|S\theta\|_2^2$. As before, this estimator can be computed exactly on a GPU with complexity $\mathcal{O}(n \log n)$.

**Proposition 3.2** ($L^2$-Fourier kernel regression). *Let $\hat{\theta}$ be the $L^2$-Fourier kernel estimator given by Equation* (2) *with $M = I$, where $I$ is the identity matrix. Then,*

$$\hat{\theta} = \mathrm{argmin}_{\theta \in \mathbb{C}^{(2m+1)^d}} n^{-1}\|\Phi\theta - \mathbb{Y}\|_2^2 + \lambda\|\theta\|_2^2$$
$$= (n^{-1}\Phi^*\Phi + \lambda I)^{-1} n^{-1}\Phi^*\mathbb{Y}.$$

*Under Assumptions (i)-(v),*

$$\mathbb{E}(\|\hat{f}_n - f^{\star}\|_{L^2(\mathbb{P}_X)}^2) \leq \inf_{\theta \in \mathbb{C}^{(2m+1)^d}} \left\{ \left(2 + 6(1+r)^2\right) \mathbb{E}(\|f^{\star} - f_{\theta}\|_{L^2(\mathbb{P}_X)}^2) + \lambda(1+r)^2\|\theta\|_2^2 \right\}$$
$$+ \lambda\sigma^2 r(1+r),$$

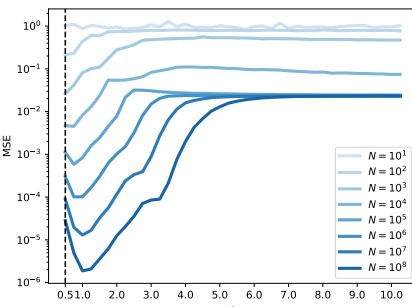 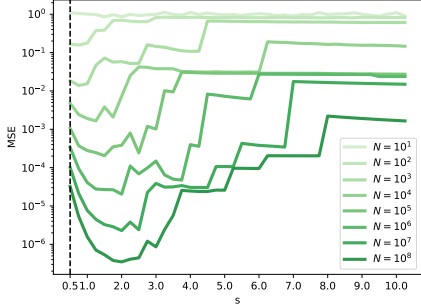

Figure 2: Sobolev regression (Left), and $L^2$-Fourier regression (Right) in dimension $d = 1$

where $r = \frac{(2m+1)^d}{\lambda n}$. *Therefore, choosing the regularization parameter as* $\lambda = n^{-2s/(2s+d)}$ *and setting the truncation level as* $m = n^{1/(2s+d)}$ *yields the convergence rate*

$$\mathbb{E}[\|f_{\hat{\theta}} - f^\star\|^2_{L^2(\mathbb{P}_X)}] = \mathcal{O}(n^{-2s/(2s+d)}).$$

The kernel associated with this estimator is know as the Dirichlet kernel (Smola & Schölkopf, 1997)

$$K(x,y) = \frac{\sin((m+1/2)(x-y)\pi/(2L))}{\sin((x-y)/(4L))}.$$

**Dependency in $s$ in the one-dimensional case.** The goal of this experiment is to compare the performance of the Sobolev kernel estimator with that of its $L^2$-Fourier (L2F) variant. We consider a one-dimensional regression setting where the input variable $X$ is uniformly distributed over the interval $\Omega = ]0, 1[$ and the response is given by $Y = 25 |X - 0.5|^3 + \varepsilon$, with a noise $\varepsilon \sim \mathcal{N}(0, 1)$ independent of $X$. We implement both kernel estimators across multiple sample sizes $n$, and evaluate their MSE on a fixed test set of $10^4$ points. To ensure a meaningful comparison, we use the same regularization and truncation schedules for both methods, setting $\lambda_n = n^{-2s/(2s+1)}$ and $m = n^{1/(2s+1)}$ for each smoothness parameter $s$. We consider 40 values of $s$ uniformly spaced between the minimal admissible value $s = d/2 = 1/2$ and $s = 10$. Figure 2 reports the MSE of the two kernel estimators, averaged over 10 independent resamples for each value of $s$. This large-scale experiment involves training 400 kernel estimators at sample size $n = 10^8$, and completes in approximately 5 minutes on a standard GPU, illustrating the computational efficiency of both kernel methods.

As expected, for any fixed value of $s$, the error of both kernel estimators decreases as the number of observations $n$ increases. In addition, we clearly observe the trade-off discussed earlier regarding the choice of the smoothness parameter $s$. Indeed, the target function $f^\star(x) = 25 |x - 0.5|^3$ belongs to the Sobolev space $H^3([0, 1])$ but not to $H^4([0, 1])$, which implies that $s = 3$ is the largest integer smoothness level for which the minimax convergence rate remains theoretically valid. However, in practice, the value $s = 3$ does not correspond to the hyperparameter that yields the lowest MSE. Interestingly, in both kernel estimators, the optimal choice of $s$ increases gradually with the sample size $n$. This empirical behavior is consistent with the theoretical results of Propositions 3.1 and 3.2, which indicate that the optimal $s$ should asymptotically converge to 3 as $n$ becomes large.

In practice, the L2F kernel consistently outperforms the Sobolev kernel. For example, at $n = 10^8$, its minimum MSE, achieved at $s = 2$, is approximately ten times lower than that of the Sobolev kernel, whose minimum occurs at $s = 1$. For the L2F kernel, the shift toward higher values of $s$ occurs more rapidly. This is due to its reduced sensitivity to the explosion of the Sobolev norm as $s$ increases, allowing it to benefit from the improved theoretical convergence rates associated with higher regularity at smaller sample sizes. Additional experiments can be found in Appendix B.1.

## 4 Physics-informed regression

Interestingly, our approach, which combines Fourier analysis with kernel methods, applies naturally to the framework of physics-informed machine learning (PIML). PIML incorporates prior physical

knowledge–typically expressed as a partial differential equation (PDE)–into regression problems in order to regularize the empirical risk (Karniadakis et al., 2021). In this context, we assume that

(vi) There is a known linear differential operator $\mathcal{D}$ with constant coefficients such that $\forall x \in \Omega$, $\mathcal{D}(f^\star, x) = 0$.

Recall that $\mathcal{D}$ is linear with constant coefficients if $\mathcal{D}(f, x) = \sum_{|\alpha| \le s} a_\alpha \, \partial^\alpha f(x)$, with $a_\alpha \in \mathbb{R}$. It turns out that this regression setting can be formulated as a kernel method on $\mathcal{H} = H^s(\Omega)$ by minimizing the empirical risk $n^{-1} \sum_{j=1}^{n} (f_\theta(X_j) - Y_j)^2 + \lambda \|f_\theta\|_{H^s(\Omega)}^2 + \mu \int_\Omega \mathcal{D}(f_\theta, x)^2 dx$, as studied in Doumèche et al. (2024); Doumèche et al. (2025a). In what follows, we show that this PDE-based penalty can be efficiently integrated into our Fourier kernel framework, where it naturally benefits from NUFFT-based acceleration.

**Tractable domains.** When the domain $\Omega$ is regular enough to allow for the analytical computation of its Fourier transform, the regression problem can be formulated equivalently on the finite-dimensional space $H_m$ as a kernel estimator, of the form:

$$\hat{\theta} = \operatorname{argmin}_{\theta \in \mathbb{C}^{(2m+1)^d}} n^{-1} \sum_{j=1}^{n} (f_\theta(X_j) - Y_j)^2 + \lambda \|f_\theta\|_{H^s}^2 + \mu \int_\Omega \mathcal{D}(f_\theta, x)^2 dx$$

$$= \operatorname{argmin}_{\theta \in \mathbb{C}^{(2m+1)^d}} n^{-1} \|\Phi\theta - \mathbb{Y}\|_2^2 + \lambda \|S\theta\|_2^2 + \mu \|C^{1/2} D\theta\|_2^2$$

$$= (n^{-1}\Phi^*\Phi + \lambda S^2 + \mu D^\star CD)^{-1} n^{-1}\Phi^*\mathbb{Y},$$

where $S$ is the Sobolev matrix of Section 3.1, the matrix $C$ is such that, for all $k_1, k_2 \in \mathbb{C}^{(2m+1)^d}$,

$$C_{k_1,k_2} = (4L)^{-d} \int_\Omega \exp(i\langle k_1 - k_2, x\rangle \pi/(2L)) dx,$$

and $D \in \mathbb{C}^{(2m+1)^d \times (2m+1)^d}$ a diagonal matrix such that $D_{k,k} = \sum_{|\alpha| \le s} a_\alpha \left(\frac{-i\pi}{2L}\right)^{|\alpha|} \prod_{\ell=1}^{d} (k_\ell)^{\alpha_\ell}$. Since $D$ is a diagonal matrix and $C$ is a $d$-level block Toeplitz and Hermitian matrix, the matrix-vector product involving $D^*CD$ can be performed in $\mathcal{O}(m^d \log m)$ operations. As a result, the conjugate gradient inversion of the full system matrix $(n^{-1}\Phi^*\Phi + \lambda S^2 + \mu D^*CD)$ can be carried out with overall complexity $\mathcal{O}(n \log n)$.

**Untractable domains.** When the Fourier decomposition of the domain $\Omega$ is not available in closed form, a common strategy in physics-informed machine learning consists in sampling a set of $n_r \in \mathbb{N}^\star$ collocation points $(X_1^{(r)}, \ldots, X_{n_r}^{(r)})$ from a known distribution $\mathbb{P}_{X^{(r)}}$ over $\Omega$. Based on these points, we then define the following kernel estimator:

$$\hat{\theta} = \operatorname{argmin}_{\theta \in \mathbb{C}^{(2m+1)^d}} n^{-1} \sum_{j=1}^{n} (f_\theta(X_j) - Y_j)^2 + \lambda \|f_\theta\|_{H^s}^2 + \mu n_r^{-1} \sum_{\ell=1}^{n_r} \mathcal{D}(f_\theta, X_j^{(r)})^2$$

$$= \operatorname{argmin}_{\theta \in \mathbb{C}^{(2m+1)^d}} n^{-1} \|\Phi\theta - \mathbb{Y}\|_2^2 + \lambda \|S\theta\|_2^2 + \mu n_r^{-1} \|\Phi D\theta\|_2^2$$

$$= (n^{-1}\Phi^*\Phi + \lambda S^2 + \mu n_r^{-1} D^*(\Phi^{(r)})^*\Phi^{(r)}D)^{-1} n^{-1}\Phi^*\mathbb{Y},$$

where $\Phi^{(r)} = (\phi(X_1^{(r)}) \mid \ldots \mid \phi(X_n^{(r)}))^*$. Since $D$ is a diagonal matrix and $(\Phi^{(r)})^*\Phi^{(r)}$ is a $d$-level block Toeplitz and Hermitian matrix, the associated matrix–vector products can be computed in $\mathcal{O}(m^d \log m)$ operations. Consequently, the conjugate gradient inversion of the system $(n^{-1}\Phi^*\Phi + \lambda S^2 + \mu n_r^{-1} D^*(\Phi^{(r)})^*\Phi^{(r)}D)$ can be performed with overall complexity $\mathcal{O}(n \log n)$.

**One dimensional example.** We consider a regression setting where the input variable $X$ is uniformly distributed over the interval $\Omega = ]0, 1[ \subseteq [-\pi/2, \pi/2]$, so that $L = \pi/2$. The response is given by $Y = \exp(X) + \varepsilon$, with a noise $\varepsilon \sim \mathcal{N}(0, 1)$ independent of $X$. The prior knowledge takes the form of the linear differential constraint $(f^\star)' - f^\star = 0$. We set the parameters as $s = d = 1$, $m = n^{1/(2s+d)} = n^{1/3}$, $\lambda = n^{-2s/(2s+d)} = n^{-2/3}$, and $\mu = 1$.

All the kernel methods process $n = 10^8$ samples in under only one second on a standard GPU (NVIDIA L4). Figure 3 reports the test error $\mathbb{E}(\|f_{\hat{\theta}} - f^\star\|_{L^2([0,1])}^2)$, evaluated on a test set of $10^4$ samples, and then averaged over 20 resamples. We see that adding physics improves the performance of the estimator, while taking approximately the same running time.

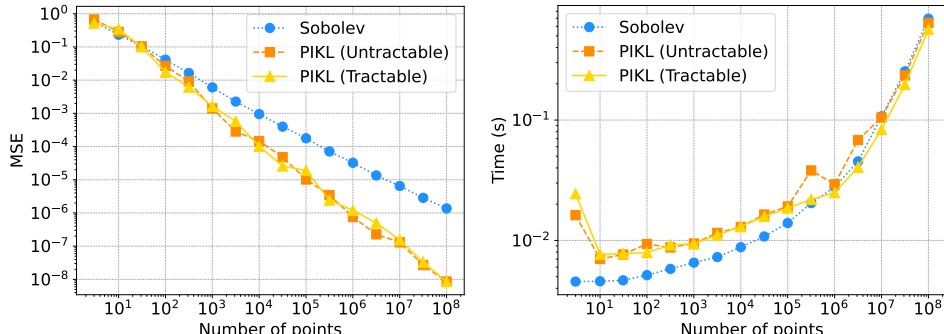

Figure 3: Physics-informed kernel regression (averaged over 20 samples)

## 5 ADDITIVE MODELS

In this section, we show how our Fourier–NUFFT framework can incorporate linear constraints on the shape of the regression function such as additivity constraints, yielding a GPU-compatible algorithm with an overall complexity of $\mathcal{O}(n \log n)$.

**Additive model.** A well-known limitation of many machine learning methods is the curse of dimensionality, reflected for instance in the minimax rate $\mathcal{O}(n^{-2s/(2s+d)})$ of the Sobolev kernel estimator of Section 3 (achieving accuracy of $\delta$ requires at least $n = \delta^{-1-d/(2s)}$ samples). To mitigate this phenomenon, a common approach is to exclude interaction effects between input variables, resulting in the following additive model:

(vii) One has $f^\star(x_1, \ldots, x_d) = g_1^\star(x_1) + \cdots + g_d^\star(x_d)$, where $g_1^\star, \ldots, g_d^\star$ are univariate functions.

The smoothness assumption on $f^\star$ naturally transfers to the functions $g_\ell^\star$. In particular $f^\star$ lies in a Sobolev space of order $s$, if and only if each function $g_\ell^\star$ belongs in a Sobolev space of order $s$.

(iii)' There exists a known smoothness parameter $s > 1/2$ such that each component function $g_1^\star, \ldots, g_d^\star$ belongs to the univariate Sobolev space of order $s$.

The $L^2$-Fourier kernel can thus be naturally extended to the additive model setting. In this case, the regression function is parametrized by a concatenated vector $\theta = (\theta_1, \ldots, \theta_d) \in \mathbb{C}^{d(2m+1)}$:

$$f_\theta(x_1, \ldots, x_d) = \langle \phi(x_1), \theta_1 \rangle + \cdots + \langle \phi(x_d), \theta_d \rangle.$$

**Proposition 5.1** (Additive kernel regression). *Let $\hat{\theta}$ be the truncated Fourier estimator defined by*

$$\hat{\theta} = \mathrm{argmin}_{\theta \in \mathbb{C}^{d(2m+1)}} n^{-1} \|\Phi_1 \theta_1 + \cdots + \Phi_d \theta_d - \mathbb{Y}\|_2^2 + \lambda \|\theta\|_2^2$$

$$= (\hat{\Sigma} + \lambda I)^{-1} (n^{-1} \Phi_1^* \mathbb{Y}, \ldots, n^{-1} \Phi_d^* \mathbb{Y})^\top$$

*where $\Phi_\ell = (\phi(X_{1,\ell}) \mid \cdots \mid \phi(X_{n,\ell}))^*$, and $\hat{\Sigma} \in \mathbb{C}^{d(2m+1) \times d(2m+1)}$ is the block-matrix such that the $(\ell_1, \ell_2)$-block is $n^{-1} \Phi_{\ell_1}^* \Phi_{\ell_2} \in \mathbb{C}^{d \times d}$. Then, under Assumptions (i), (ii), (iii)', (iv), (v), and (vii), choosing $\lambda = n^{-2s/(2s+1)}$ and $m = n^{1/(2s+1)}/d$ yields the convergence rate*

$$\mathbb{E}(\|f_{\hat{\theta}} - f^\star\|_{L^2(\mathbb{P}_X)}^2) = \mathcal{O}(n^{-2s/(2s+1)}).$$

The kernel associated with this estimator is the sum Dirichlet kernel

$$K_d((x_1, \ldots, x_d), (y_1, \ldots, y_d)) = \sum_{j=1}^{d} \frac{\sin((m + 1/2)(x_j - y_j)\pi/(2L))}{\sin((x_j - y_j)/(4L))}. \tag{3}$$

Notice that the additive assumption allows us to recover the more favorable univariate minimax rate of $n^{-2s/(2s+1)}$, as in the case $d = 1$. The scaling $m = n^{1/(2s+1)}/d$ is consistent with existing results on generalized additive models (GAMs) in high dimensions (Wood et al., 2014, Section 3.3). We provide in Appendix A.2 an explicit error bound for the additive kernel regression.

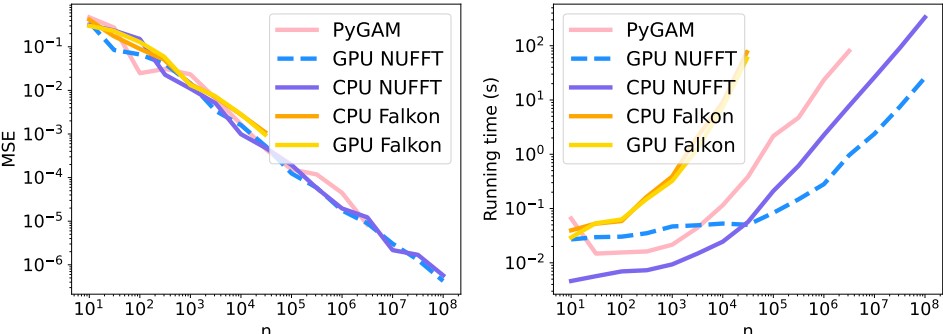

Figure 4: Additive model with $d = 5$

**Complexity.** Key components of the matrix $\hat{\Sigma}$—both diagonal and off-diagonal blocks—can be efficiently computed using NUFFT in $\mathcal{O}(d^2 n \log n)$ time, enabling the full system to be solved in nearly linear time when $s \geq 1$ (see Appendix A.2 for details) with GPU acceleration. This is significantly more efficient than GAMs, which require $\mathcal{O}(d^2 n m^2)$ operations and cannot benefit from NUFFT acceleration due to their reliance on spline bases rather than Fourier representations (Wood et al., 2014). In addition, the memory footprint of our kernel method is $\mathcal{O}(n \log n + m^2 d^2) = \mathcal{O}(n \log n)$, while GAMs require $\mathcal{O}(ndm)$ memory to store the design matrix.

**Simple additive model without grid search** In this example, we consider the following additive regression setting in dimension $d = 5$:

(i) The input variable $X$ is uniformly distributed over the hypercube $\Omega = ]0, 1[^5$;

(ii) The response is given by

$$Y = \sum_{\ell=1}^{5} \Big( \exp \big( \frac{X_\ell}{\ell + 1} \big) - 1 \Big) + \varepsilon,$$

where $\varepsilon \sim \mathcal{N}(0, 1)$ is independent of $X$.

We implement the $L^2$-Fourier additive kernel regression estimator with smoothness parameter $s = 2$, and evaluate its performance over 16 different values of $n$ ranging from 1 to $10^8$. In Figure 4, we compare our $L^2$-Fourier additive kernel estimator implemented on CPU (CPU NUFFT) and on a NVIDIA T4 GPU (GPU NUFFT) with the `PyGAM` package, which runs on CPU only (Servén & Brummitt, 2018), and the `FALKON` Python package for both CPU and GPU (Rudi et al., 2017). For a fair comparison, the `PyGAM` estimator is given $2m + 1$ splines for each of the five univariate components, where $m = 1 + \lfloor n^{-1/(2s+1)}/d \rfloor$ with $s = 2$. The `FALKON` algorithm is implemented using Equation (3). This ensures that both kernels and the spline-based estimators use the same number of parameters. In all methods, the regularization parameter is set to $\lambda_n = n^{-2s/(2s+1)}$. Figure 4 (Left) shows that all estimators achieve comparable predictive performance. Figure 4 (Right) highlights the computational advantages of our method. The NUFFT kernel estimators exhibit significantly faster runtimes compared to `PyGAM` and `FALKON`, especially as $n$ increases. In particular, `PyGAM` exceeds the available CPU memory (15 GB) at $n = 10^7$, making it infeasible to evaluate for larger datasets. In contrast, our GPU implementation remains scalable and efficient. For instance, at $n = 10^8$, the GPU kernel estimator is approximately 14 times faster than its CPU counterpart. The GPU implementation of `FALKON` only speed up the computation by approximately 15%.

**Grid search for $\lambda$.** Interestingly, the most computationally demanding step in the kernel estimation pipeline is the NUFFT computation, which has a complexity of $\mathcal{O}(d^2 n \log n)$. In contrast, the subsequent matrix inversion step is significantly faster, with a complexity of $\mathcal{O}(d^3 m^3) = \mathcal{O}(d^3 n^{-3/5})$ when $s = 2$. This observation is particularly relevant because it implies that multiple values of the regularization parameter $\lambda$ can be tested efficiently, making grid search highly tractable. Figure 5 compares the `gam.gridsearch` function from `PyGAM` to our kernel method

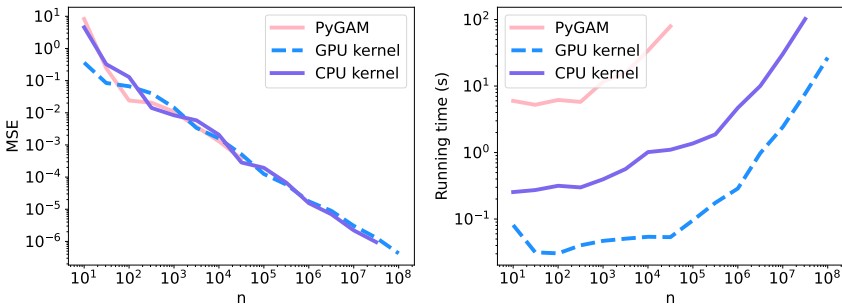

Figure 5: Grid search with 300 hyperparameters for the additive model with $d = 5$

implemented on both CPU and GPU, using a grid of 300 candidate values for $\lambda$. While the runtime of the `PyGAM` grid search becomes prohibitive at $n = 10^5$, our GPU-based kernel implementation completes the entire grid search in under 30 seconds even for $n = 10^8$.

## CONCLUSION

In this work, we have introduced a scalable framework for kernel methods with $\mathcal{O}(n \log n)$ complexity, designed to fully exploit GPU acceleration. The approach combines a Fourier representation of the kernel with the non-uniform fast Fourier transform (NUFFT), yielding fast and memory-efficient computations. We instantiate the framework for Sobolev, physics-informed, and additive kernel models, thereby demonstrating its flexibility and broad applicability. From a theoretical perspective, we establish that the proposed Sobolev and additive kernels achieve optimal minimax convergence rates, in line with classical results from kernel theory.

Future works could focus on speeding up the hyperparameter search by using tailored preconditioners for the conjugate gradient algorithm. The hyperparameter tuning could also be quickened by implementing gradient descent algorithm leveraging the fact that the gradient of the empirical risk admits a closed-from solution, similarly to what is done in Greengard (2025).

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

# A ADDITIONAL RESULTS

## A.1 PROPERTIES OF FOURIER EXPANSION FOR SOBOLEV FUNCTIONS

The smoothness of a function in $H^s(\Omega)$ (endowed with any of the equivalent Sobolev norms of order $s$) is reflected in the decay rate of its Fourier coefficients, as formally stated in the next result.

**Proposition A.1** (Fourier series representation, Doumèche et al., 2025a). *Let $s \in \mathbb{N}^\star$. There exists a constant $C_\Omega > 0$ such that, for any function $f \in H^s(\Omega)$, there is a sequence of Fourier coefficients $\theta(f) \in \mathbb{C}^{\mathbb{Z}^d}$ such that:*

*(i)* $f(x) = \langle \phi_\infty(x), \theta(f) \rangle$, *where* $\phi_\infty(x) = \left( \exp\left(-i\frac{\pi}{2L}\langle k, x \rangle\right) \right)_{k \in \mathbb{Z}^d}$*;*

*(ii) The following norm equivalence holds:*

$$\|f\|_{H^s(\Omega)}^2 \leq \sum_{k \in \mathbb{Z}^d} |\theta(f)_k|^2 \left(1 + \|k\|_2^{2s}\right) \leq C_\Omega \|f\|_{H^s(\Omega)}^2.$$

## A.2 ADDITIONAL ELEMENTS FOR ADDITIVE MODELS

In this section, we provide an explicit upper bound on the error of an additive kernel regressor depending on the regularization strength and the truncation level.

**Proposition A.2** (Additive kernel regression). *Let $\hat{\theta}$ be the truncated Fourier estimator defined by*

$$\hat{\theta} = \operatorname{argmin}_{\theta \in \mathbb{C}^{d(2m+1)}} n^{-1} \|\Phi_1 \theta_1 + \cdots + \Phi_d \theta_d - \mathbb{Y}\|_2^2 + \lambda \|\theta\|_2^2$$

$$= (\hat{\Sigma} + \lambda I)^{-1} (n^{-1}\Phi_1^* \mathbb{Y}, \ldots, n^{-1}\Phi_d^\star \mathbb{Y})^\top$$

*where $\Phi_\ell = (\phi(X_{1,\ell}) \mid \cdots \mid \phi(X_{n,\ell}))^\star$, $I$ is the $d(2m+1) \times d(2m+1)$ identity matrix, and $\hat{\Sigma}$ is the block-matrix such that the $(\ell_1, \ell_2)$-block is $n^{-1}\Phi_{\ell_1}^* \Phi_{\ell_2}$, i.e.,*

$$\hat{\Sigma} = \begin{pmatrix} n^{-1}\Phi_1^*\Phi_1 & \ldots & n^{-1}\Phi_1^*\Phi_d \\ \vdots & \ddots & \vdots \\ n^{-1}\Phi_d^*\Phi_1 & \ldots & n^{-1}\Phi_d^*\Phi_d \end{pmatrix}.$$

*Then, under Assumptions (i), (ii), (iii)', (iv), (v), and (vii), one has*

$$\mathbb{E}(\|\hat{f}_n - f^\star\|_{L^2(\mathbb{P}_X)}^2) \leq \inf_{\theta \in \mathbb{C}^{d(2m+1)}} \left( (2 + 6(1+r)^2)\mathbb{E}(\|f^\star - f_\theta\|_{L^2(\mathbb{P}_X)}^2) + \lambda(1+r)^2\|\theta\|_2^2 \right)$$
$$+ \lambda\sigma^2 r(1+r),$$

*where $r = \frac{d(2m+1)}{\lambda n}$. Therefore, choosing the regularization parameter as $\lambda = n^{-2s/(2s+1)}$ and setting the truncation level as $m = n^{1/(2s+1)}/d$ yields the convergence rate*

$$\mathbb{E}(\|f_{\hat{\theta}} - f^\star\|_{L^2(\mathbb{P}_X)}^2) = \mathcal{O}(n^{-2s/(2s+1)}).$$

**Complexity.** In this part, we discuss an efficient way to implement $\hat{\theta}$. As explained in Section 2.3, the terms $n^{-1}\Phi_1^*\mathbb{Y}$ and the diagonal blocks $n^{-1}\Phi_\ell^*\Phi_\ell$ of the matrix $\hat{\Sigma}$ can be efficiently computed using the NUFFT. As a result, the vector $n^{-1}(\Phi_1^*\mathbb{Y}, \ldots, \Phi_d^*\mathbb{Y})^\top$ and the diagonal of $\hat{\Sigma}$ can be computed with overall complexity $\mathcal{O}(dn \log n)$. Moreover, the off-diagonal block $n^{-1}\Phi_{\ell_1}^*\Phi_{\ell_2}$ corresponds to the 2d-NUFFT of the constant function equal to 1, evaluated at the points $(X_{1,\ell_1}, -X_{1,\ell_2}), \ldots, (X_{n,\ell_1}, -X_{n,\ell_2})$. The matrix $n^{-1}\Phi_{\ell_1}^*\Phi_{\ell_2}$ can thus be computed in $\mathcal{O}(n \log n)$. Consequently, the full matrix $\hat{\Sigma}$ can be assembled in $\mathcal{O}(d^2 n \log n)$ operations (since there are $d^2$ blocks of $\Phi^\star\Phi$-type). The linear system involving $(\hat{\Sigma} + \lambda I)$ can then be solved in $\mathcal{O}(d^3 m^3) = \mathcal{O}(n)$ operations, provided that $s \geq 1$.

# B ADDITIONAL EXPERIMENTS

## B.1 DEPENDENCY IN $s$ IN THE TWO-DIMENSIONAL CASE FOR THE SOBOLEV REGRESSION.

In this experiment, we aim to compare the performance of the Sobolev kernels in dimension 2, for the task of approximating smooth functions. The regression setting is defined as follows:

(i) The input $X = (X^{(1)}, X^{(2)})$ is uniformly distributed over the square domain $\Omega = ]0, 1[^2$;

(ii) The response is given by $Y = \exp(X^{(1)}) \cos(X^{(2)}) + \varepsilon$, where the noise term $\varepsilon \sim \mathcal{N}(0, 1)$ is independent of $X$.

We implement both kernel estimators for several sample sizes $n$, and evaluate their MSE on a test set of $10^4$ points. To ensure a meaningful comparison, we adopt the same regularization and truncation schedules for both methods, setting $\lambda_n = n^{-2s/(2s+2)}$ and $m = n^{1/(2s+2)}$. We consider 40 values of the smoothness parameter $s$, uniformly spaced between $s = d/2 = 1$ and $s = 10$. Figure 6 reports the MSEs of both techniques, averaged over 10 independent resamples for each value of $s$. This large-scale experiment, which involves training 400 kernel estimators at $n = 10^8$, completes in approximately 21 minutes on a standard GPU (NVIDIA L4).

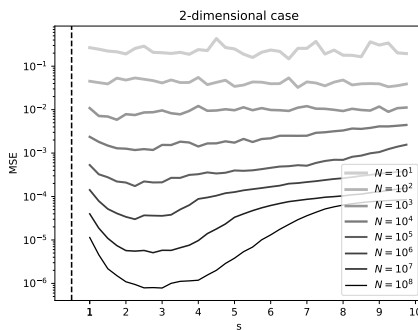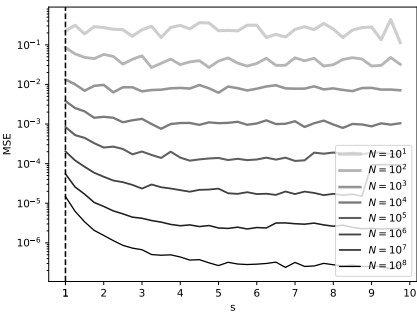

Figure 6: Sobolev regression (Left), and $L^2$-Fourier regression (Right) in dimension $d = 2$

Once again, the $L^2$-Fourier kernel consistently outperforms the Sobolev kernel. In this example, the target function satisfies $f^\star \in \cap_{s \in \mathbb{N}} H^s(\Omega)$, meaning that it is infinitely smooth and the theoretically optimal regularity parameter is $s^\star = \infty$. As in the one-dimensional case, the optimal value of $s$ for the Sobolev kernel increases more slowly toward $s^\star$ than for the $L^2$-Fourier estimator, which benefits more rapidly from the higher smoothness of $f^\star$.

## C    PROOFS

### C.1    PROOF OF PROPOSITION 2.1

For $f \in H^s(\Omega)$, let $\theta(f) \in \mathbb{C}^{\mathbb{Z}^d}$ be the Fourier coefficients of $f$, as in Proposition A.1. To simplify notation, we write $\theta = \theta(f)$. Let $f_m$ be the projection of $f$ on $H_m$, defined by

$$f_m(x) = \sum_{\|k\|_\infty \leq m} \theta_k \exp(i\pi \langle k, x \rangle / 2L).$$

Parseval's identity ensures that

$$\mathbb{E}((f(X) - f_m(X))^2) \leq \kappa \int_{[-2L, 2L]^d} (f(x) - f_m(x))^2 dx = \kappa (4L)^d \sum_{\|k\|_\infty > m} |\theta_k|^2. \qquad (4)$$

Since $f \in H^s(\Omega)$, one has $\sum_{k \in \mathbb{Z}^d} |\theta_k|^2 \|k\|_2^{2s} < \infty$. Let $k(\ell)$ be a reindexing of $\mathbb{Z}^d$ such that $\|k(\ell)\|_2$ is non-decreasing. According to Doumèche et al. (2024, Proposition A.7), there is a constant $C_1 > 0$ such that $\|k(\ell)\|_2^{2s} \leq C_1 \ell^{2s/d}$. Moreover, since $\|k\|_\infty > m \Rightarrow \|k\|_2 > m$, we deduce that

$$\sum_{\|k\|_\infty > m} |\theta_k|^2 = \sum_{\|k\|_\infty > m} (|\theta_k| \|k\|_2^{2s}) \|k\|_2^{-2s} \leq \sum_{\ell > (2m+1)^d} (|\theta_{k(\ell)}|^2 \|k(\ell)\|_2^{2s}) \|k(\ell)\|_2^{-2s}.$$

Now, consider the Abel transform $A_\ell = \sum_{j=0}^{\ell-1} (|\theta_{k(j)}|^2 \|k(j)\|_2^{2s})$ of the sum above. By construction, $A_\ell$ is bounded by $C_\Omega \|f\|_{H^s(\Omega)}^2$. Note that

$$\sum_{\ell > (2m+1)^d} (|\theta_{k(\ell)}|^2 \|k(\ell)\|_2^{2s}) \|k(\ell)\|_2^{-2s} = \sum_{\ell > (2m+1)^d} (A_{\ell+1} - A_\ell) \|k(\ell)\|_2^{-2s}.$$

This telescopic series satisfies the Abel property (discrete version of the integration by parts), i.e.,

$$\sum_{\ell > (2m+1)^d} (A_{\ell+1} - A_\ell) \|k(\ell)\|_2^{-2s}$$

$$= A_{(2m+1)^d} \|k((2m+1)^d)\|_2^{-2s} + \sum_{\ell > (2m+1)^d} A_{\ell+1} (\|k(\ell)\|_2^{-2s} - \|k(\ell+1)\|_2^{-2s}).$$

Moreover, since $\|k(\ell)\|_2$ is non-decreasing, the right-hand term is bounded by

$$\sum_{\ell > (2m+1)^d} A_{\ell+1}(\|k(\ell)\|_2^{-2s} - \|k(\ell+1)\|_2^{-2s})$$

$$\leq C_\Omega \|f\|_{H^s(\Omega)}^2 \sum_{\ell > (2m+1)^d} (\|k(\ell)\|_2^{-2s} - \|k(\ell+1)\|_2^{-2s})$$

$$= C_\Omega \|f\|_{H^s(\Omega)}^2 \|k((2m+1)^d)\|_2^{-2s}.$$

Overall, we deduce that

$$\sum_{\|k\|_\infty > m} |\theta_k|^2 \leq 2C_\Omega \|f\|_{H^s(\Omega)}^2 \|k((2m+1)^d)\|_2^{-2s}$$

$$\leq 2C_1 C_\Omega \|f\|_{H^s(\Omega)}^2 (2m+1)^{-2sd/d}$$

$$\leq 2C_1 C_\Omega \|f\|_{H^s(\Omega)}^2 m^{-2s}.$$

Combining this result with (4), we deduce that

$$\mathbb{E}((f(X) - f_m(X))^2) \leq C\kappa \|f\|_{H^s(\Omega)}^2 m^{-2s},$$

where $C = 2C_1 C_\Omega (4L)^d$.

### C.2   Proof of Propositions 3.1 and 3.2

**General kernel regression setting.**   Let $\theta \in \mathbb{C}^{(2m+1)^d}$ be an arbitrary vector of Fourier coefficients. We are interested in bounding the quantity

$$\mathbb{E}(\|f_{\hat{\theta}} - f^\star\|_{L^2(\mathbb{P}_X)}^2) = \mathbb{E}(\|(f_{\hat{\theta}} - f_\theta) + (f_\theta - f^\star)\|_{L^2(\mathbb{P}_X)}^2)$$

$$\leq 2\mathbb{E}(\|f_{\hat{\theta}} - f_\theta\|_{L^2(\mathbb{P}_X)}^2) + 2\mathbb{E}(\|f_\theta - f^\star\|_{L^2(\mathbb{P}_X)}^2). \tag{5}$$

Let us start by bounding the error term $\mathbb{E}(\|f_{\hat{\theta}} - f_\theta\|_{L^2(\mathbb{P}_X)}^2)$.

**Generalization error.**   In both propositions, $\hat{\theta}$ can be written as

$$\hat{\theta} = (\hat{\Sigma} + \lambda R)^{-1} \Phi^\star \mathbb{Y}/n,$$

where $\hat{\Sigma} = n^{-1}\Phi^\star \Phi$ is the empirical covariance matrix and $R$ is a regularization matrix: $R = S^2$ in the Sobolev regression, and $R = I$ for the $L^2$-Fourier regression. Let $\Sigma = \mathbb{E}(\hat{\Sigma})$ be the theoretical covariance matrix, defined by $\langle \theta_1, \Sigma\theta_2 \rangle = \mathbb{E}(\langle \phi(X), \theta_1 \rangle \langle \phi(X), \theta_2 \rangle)$. Since $\mathbb{Y}$ can be decomposed as $\mathbb{Y} = \Phi\theta + \delta\mathbb{Y} + \epsilon$, where $\epsilon = (\varepsilon_1, \ldots, \varepsilon_n)^\top$ and $\delta\mathbb{Y}_j = f^\star(X_j) - f_\theta(X_j)$, we can write

$$\hat{\theta} = (\hat{\Sigma} + \lambda R)^{-1} \Phi^\star \mathbb{Y}/n$$

$$= (\hat{\Sigma} + \lambda R)^{-1} (\hat{\Sigma}\theta + \Phi^\star(\delta\mathbb{Y} + \epsilon)/n)$$

$$= \theta + (\hat{\Sigma} + \lambda R)^{-1} (-\lambda R\theta + \Phi^\star(\delta\mathbb{Y} + \epsilon)/n).$$

This leads to the following decomposition of the generalization error:

$$\mathbb{E}(\|f_{\hat{\theta}} - f_\theta\|_{L^2(\mathbb{P}_X)}^2) = \mathbb{E}(\|\sqrt{\Sigma}(\hat{\theta} - \theta)\|_2^2)$$

$$= \mathbb{E}(\|\sqrt{\Sigma}(\hat{\Sigma} + \lambda R)^{-1}(-\lambda R\theta + \Phi^*(\epsilon + \delta\mathbb{Y})/n)\|_2^2)$$

$$\leq 3\mathbb{E}(\|\sqrt{\Sigma}(\hat{\Sigma} + \lambda R)^{-1}\lambda R\theta\|_2^2) + 3\mathbb{E}(\|\sqrt{\Sigma}(\hat{\Sigma} + \lambda R)^{-1}\Phi^*\epsilon/n\|_2^2)$$

$$+ 3\mathbb{E}(\|\sqrt{\Sigma}(\hat{\Sigma} + \lambda R)^{-1}\Phi^*\delta\mathbb{Y}/n\|_2^2),$$

where we used the fact that $(x_1 + x_2 + x_3)^2 \leq 3x_1^2 + 3x_2^2 + 3x_3^2$. The first term $\mathbb{E}(\|\sqrt{\Sigma}(\hat{\Sigma} + \lambda R)^{-1}\lambda R\theta\|_2^2)$ is a bias term depending on the regularization $\lambda R$. The second term $\mathbb{E}(\|\sqrt{\Sigma}(\hat{\Sigma} + \lambda R)^{-1}\Phi^*\epsilon/n\|_2^2)$ is a variance term depending on the noise $\epsilon$. The last term $\mathbb{E}(\|\sqrt{\Sigma}(\hat{\Sigma} + \lambda R)^{-1}\Phi^*\delta\mathbb{Y}/n\|_2^2)$ is another approximation error term, which measures the impact of the interaction between the regularization $R$ and the approximation errors $\delta\mathbb{Y}$. In what follows, we will successively bound these three errors terms.

**Bias term.** According to Bach (2024, Lemma 7.1), one has

$$\mathbb{E}(\|\sqrt{\Sigma}(\hat{\Sigma} + \lambda R)^{-1}\lambda R\theta\|_2^2) = \lambda^2 \mathbb{E}(\|\sqrt{\Sigma}R^{-1/2}(R^{-1/2}\hat{\Sigma}R^{-1/2} + \lambda)^{-1}R^{1/2}\theta\|_2^2)$$

$$\leq \lambda(1 + \frac{\alpha^2}{\lambda n})^2 \langle R^{1/2}\theta, R^{-1/2}\Sigma R^{-1/2}(R^{-1/2}\Sigma R^{-1/2} + \lambda)^{-1}R^{1/2}\theta \rangle$$

$$\leq \lambda(1 + \frac{\alpha^2}{\lambda n})^2 \|R^{1/2}\theta\|_2^2,$$

where $\alpha = \max_{x \in \mathbb{R}^d} \|R^{-1/2}\phi(x)\|_2$. Observe that $R^{1/2} = S$ for the Sobolev regression and $R^{1/2} = I$ for the $L^2$-Fourier regression. Thus, $\alpha^2 = \sum_{k \in \mathbb{Z}^d} \frac{1}{1+\|k\|_2^{2s}} < \infty$ for the Sobolev regression, while $\alpha^2 = (2m+1)^d$ for the $L^2$-Fourier regression. Remark that $\|R^{1/2}\theta\|_2^2$ is the kernel norm.

**Variance term.** By expanding $\epsilon = \sum_{j=1}^n \varepsilon_j e_j$ on the canonical basis $(e_1, \ldots, e_n)$, we have that

$$\mathbb{E}(\|\sqrt{\Sigma}(\hat{\Sigma} + \lambda R)^{-1}\Phi^*\epsilon/n)\|_2^2)$$

$$= n^{-2} \sum_{j_1, j_2 = 1}^n \mathbb{E}(e_{j_1}^\star \Phi(\hat{\Sigma} + \lambda R)^{-1}\Sigma(\hat{\Sigma} + \lambda R)^{-1}\Phi^* e_{j_2} \mathbb{E}(\varepsilon_{j_1}\varepsilon_{j_2} \mid X_1, \ldots X_n)).$$

Since $\varepsilon_{j_1}$ and $\varepsilon_{j_2}$ are independent when $j_1 \neq j_2$, and since $\mathbb{E}(\varepsilon_j \mid X_1, \ldots X_n)) = 0$, we deduce that

$$\mathbb{E}(\|\sqrt{\Sigma}(\hat{\Sigma} + \lambda R)^{-1}\Phi^*\varepsilon/n\|_2^2)$$

$$= \mathbb{E}(\varepsilon_1^2) n^{-2} \text{tr}(\mathbb{E}(\Phi(\hat{\Sigma} + \lambda R)^{-1}\Sigma(\hat{\Sigma} + \lambda R)^{-1}\Phi^*))$$

$$\leq \sigma^2 n^{-2} \text{tr}(\mathbb{E}(\Phi R^{-1/2}(R^{-1/2}\hat{\Sigma}R^{-1/2} + \lambda)^{-1}R^{-1/2}\Sigma R^{-1/2}(R^{-1/2}\hat{\Sigma}R^{-1/2} + \lambda)^{-1}R^{-1/2}\Phi^*))$$

$$= \sigma^2 n^{-1} \text{tr}(\mathbb{E}((R^{-1/2}\hat{\Sigma}R^{-1/2} + \lambda)^{-1}R^{-1/2}\Sigma R^{-1/2}(R^{-1/2}\hat{\Sigma}R^{-1/2} + \lambda)^{-1}R^{-1/2}\hat{\Sigma}R^{-1/2}))$$

$$\leq \sigma^2 n^{-1} \text{tr}(\mathbb{E}((R^{-1/2}\hat{\Sigma}R^{-1/2} + \lambda)^{-1}R^{-1/2}\Sigma R^{-1/2})).$$

Using Bach (2024, Lemma 7.1), we conclude that

$$\mathbb{E}(\|\sqrt{\Sigma}(\hat{\Sigma} + \lambda R)^{-1}\Phi^*\varepsilon/n\|_2^2) \leq \sigma^2 n^{-1}(1 + \frac{\alpha^2}{\lambda n})\text{tr}(\mathbb{E}((R^{-1/2}\Sigma R^{-1/2} + \lambda)^{-1}R^{-1/2}\Sigma R^{-1/2}))$$

$$\leq \sigma^2 n^{-1}(1 + \frac{\alpha^2}{\lambda n})(2m+1)^d.$$

**Approximation term.** Let $(X_{n+1}, Y_{n+1})$ be a new sample, drawn independently from $(X_1, Y_1), \ldots, (X_n, Y_n)$. Let

$$C = \sum_{j=1}^{n+1} R^{-1/2}\phi(X_j)\phi(X_j)^* R^{-1/2},$$

so that $C = nR^{-1/2}\hat{\Sigma}R^{-1/2} + R^{-1/2}\phi(X_{n+1})\phi(X_{n+1})^* R^{-1/2}$. The rationale behind this approach is that $(X_{n+1}, Y_{n+1})$ relates to $\Sigma$ by $R^{-1/2}\Sigma R^{-1/2} = \mathbb{E}(R^{-1/2}\phi(X_{n+1})\phi(X_{n+1})^* R^{-1/2})$. Moreover, $\hat{\Sigma}$ and $C$ are related by Bach (2024, Equation (7.26)), so that

$$(C + \lambda n)^{-1}R^{-1/2}\phi(X_{n+1})$$

$$= (1 + \langle R^{-1/2}\phi(X_{n+1}), (nR^{-1/2}\hat{\Sigma}R^{-1/2} + n\lambda)^{-1}R^{-1/2}\phi(X_{n+1})\rangle)^{-1}$$

$$\times (nR^{-1/2}\hat{\Sigma}R^{-1/2} + n\lambda)^{-1}R^{-1/2}\phi(X_{n+1}),$$

where the scaling coefficient is such that

$$(1 + \langle R^{-1/2}\phi(X_{n+1}), (nR^{-1/2}\hat{\Sigma}R^{-1/2} + n\lambda)^{-1}R^{-1/2}\phi(X_{n+1})\rangle)^{-1} \geq (1 + \frac{\alpha^2}{\lambda n})^{-1}.$$

Thus,

$$\mathbb{E}(\|\sqrt{\Sigma}(\hat{\Sigma} + \lambda R)^{-1}\Phi^* \delta \mathbb{Y}/n\|_2^2)$$

$$= n^{-2}\mathbb{E}((\delta \mathbb{Y})^{\star}\Phi(\hat{\Sigma} + \lambda R)^{-1}\phi(X_{n+1})\phi(X_{n+1})^*(\hat{\Sigma} + \lambda R)^{-1}\Phi^* \delta \mathbb{Y})$$

$$= n^{-2}\mathbb{E}(|\langle \Phi(\hat{\Sigma} + \lambda R)^{-1}\phi(X_{n+1}), \delta \mathbb{Y}\rangle|^2)$$

$$= n^{-2}\mathbb{E}(|\langle \Phi R^{-1/2}(R^{-1/2}\hat{\Sigma}R^{-1/2} + \lambda)^{-1}R^{-1/2}\phi(X_{n+1}), \delta \mathbb{Y}\rangle|^2)$$

$$\leq (1 + \frac{\alpha^2}{\lambda n})^2 \mathbb{E}(|\langle \Phi R^{-1/2}(C + \lambda n)^{-1}R^{-1/2}\phi(X_{n+1}), \delta \mathbb{Y}\rangle|^2)$$

$$\leq (1 + \frac{\alpha^2}{\lambda n})^2 \mathbb{E}(\|\Phi R^{-1/2}(C + \lambda n)^{-1}R^{-1/2}\phi(X_{n+1})\|_2^2 \, \|\delta \mathbb{Y}\|^2).$$

Since $R^{-1/2}\Phi^{\star}\Phi R^{-1/2} = nR^{-1/2}\hat{\Sigma}R^{-1/2} \leq C$, we deduce that

$$\|\Phi R^{-1/2}(C + \lambda n)^{-1}R^{-1/2}\phi(X_{n+1})\|_2^2$$

$$\leq \phi(X_{n+1})^* R^{-1/2}(C + \lambda n)^{-1}C(C + \lambda n)^{-1}R^{-1/2}\phi(X_{n+1})$$

$$\leq \phi(X_{n+1})^* R^{-1/2}(C + \lambda n)^{-1}R^{-1/2}\phi(X_{n+1}).$$

Thus,

$$\mathbb{E}(\|\sqrt{\Sigma}(\hat{\Sigma} + \lambda R)^{-1}\Phi^* \delta \mathbb{Y}/n\|_2^2)$$

$$\leq (1 + \frac{\alpha^2}{\lambda n})^2 \mathbb{E}(\|(C + \lambda n)^{-1/2}R^{-1/2}\phi(X_{n+1})\|_2^2 \, \|\delta \mathbb{Y}\|^2)$$

$$\leq (1 + \frac{\alpha^2}{\lambda n})^2 \mathbb{E}(\|(C + \lambda n)^{-1/2}R^{-1/2}\phi(X_{n+1})\|_2^2 \, (\|\delta \mathbb{Y}\|^2 + (\delta Y_{n+1})^2)).$$

Since the observations $(X_1, Y_1), \ldots, (X_{n+1}, Y_{n+1})$ are assumed i.i.d., the random variables $\|(C + \lambda n)^{-1/2}R^{-1/2}\phi(X_j)\|_2^2 \, (\|\delta \mathbb{Y}\|^2 + (\delta Y_{n+1})^2)$ have the same distribution for all $1 \leq j \leq n + 1$. Therefore, they also share the same expectation. Thus,

$$\mathbb{E}(\|(C + \lambda n)^{-1/2}R^{-1/2}\phi(X_{n+1})\|_2^2 \, (\|\delta \mathbb{Y}\|^2 + (\delta Y_{n+1})^2))$$

$$= \frac{1}{n+1}\mathbb{E}\Big(\sum_{j=1}^{n+1}\|(C + \lambda n)^{-1/2}R^{-1/2}\phi(X_j)\|_2^2 \, (\|\delta \mathbb{Y}\|^2 + (\delta Y_{n+1})^2)\Big)$$

$$= \frac{1}{n+1}\mathbb{E}\Big(\sum_{j=1}^{n+1}\text{tr}(\phi(X_j)^* R^{-1/2}(C + \lambda n)^{-1}R^{-1/2}\phi(X_j)) \, (\|\delta \mathbb{Y}\|^2 + (\delta Y_{n+1})^2)\Big)$$

$$= \frac{1}{n+1}\mathbb{E}\Big(\text{tr}((C + \lambda n)^{-1}\sum_{j=1}^{n+1}R^{-1/2}\phi(X_j)\phi(X_j)^* R^{-1/2}) \, (\|\delta \mathbb{Y}\|^2 + (\delta Y_{n+1})^2)\Big)$$

$$= \frac{1}{n+1}\mathbb{E}\Big(\text{tr}((C + \lambda n)^{-1}C) \, (\|\delta \mathbb{Y}\|^2 + (\delta Y_{n+1})^2)\Big)$$

$$\leq \frac{1}{n+1}\mathbb{E}\Big((\|\delta \mathbb{Y}\|^2 + (\delta Y_{n+1})^2)\Big) = \mathbb{E}((\delta Y^2)) = \mathbb{E}(\|f^{\star} - f_\theta\|_{L^2(\mathbb{P}_X)}^2),$$

where we use the fact $x = \text{tr}(x)$ for any real number $x$, and $\text{tr}(M_1 M_2) = \text{tr}(M_2 M_1)$. All in all, we have

$$\mathbb{E}(\|\sqrt{\Sigma}(\hat{\Sigma} + \lambda R)^{-1}\Phi^* \delta \mathbb{Y}/n\|_2^2) \leq (1 + \frac{\alpha^2}{\lambda n})^2 \, \mathbb{E}(\|f^{\star} - f_\theta\|_{L^2(\mathbb{P}_X)}^2).$$

**Conclusion.** Putting everything together, we deduce from inequality (5) that

$$\mathbb{E}(\|f_{\hat{\theta}} - f^{\star}\|_{L^2(\mathbb{P}_X)}^2) \leq \inf_{\theta \in H_m} \Big((2 + 6(1 + \frac{\alpha^2}{\lambda n})^2)\mathbb{E}(\|f^{\star} - f_\theta\|_{L^2(\mathbb{P}_X)}^2) + \lambda(1 + \frac{\alpha^2}{\lambda n})^2\|R^{1/2}\theta\|_2^2\Big)$$

$$+ \sigma^2 n^{-1}(1 + \frac{\alpha^2}{\lambda n})(2m + 1)^d,$$

where

- $\alpha^2 = \sum_{k \in \mathbb{Z}^d} \frac{1}{1 + \|k\|_2^{2s}} < \infty$ for the Sobolev regression, while $\alpha^2 = (2m+1)^d$ for the $L^2$-Fourier regression;

- $\|R^{1/2}\theta\|_2^2 = \|S\theta\|_2^2 \le C_\Omega \|f_\theta\|_{H^s(\Omega)}^2$ for the Sobolev regression, and $\|R^{1/2}\theta\|_2^2 = \|\theta\|_2^2$ for the $L^2$-Fourier regression.

**Convergence rates.** According to Proposition A.1, there is a parameter $\theta(f^\star)$ such that $f^\star(x) = \langle \phi(x), \theta(f^\star) \rangle$. Let $\theta^\star \in \mathbb{C}^{(2m+1)^d}$ be the truncation of $\theta(f^\star)$, i.e., $\theta_k^\star = \theta(f^\star)_k$. Proposition 2.1 states that the approximation error term $\mathbb{E}(\|f_{\theta^\star} - f^\star\|_{L^2(\mathbb{P}_X)}^2) = \mathcal{O}(n^{-\gamma})$, where $\gamma = 2s/(2s+d)$. Moreover, the RKHS norm of $\theta^\star$ is bounded by $\max(\|\theta^\star\|_2^2, \|S\theta^\star\|_2^2) \le \sum_{k \in \mathbb{C}^{\mathbb{Z}^d}} (1 + \|k\|_2^{2s}) |\theta(f^\star)_k|^2 \le C_\Omega \|f^\star\|_{H^s(\Omega)}^2$. Therefore, in the Sobolev regression, setting $\lambda = \mathcal{O}(n^{-2s/(2s+d)})$ such that $\lambda \ge n^{-1}$, and taking $m = n^{1/(2s+d)}$ leads to Sobolev minimax rate. Moreover, in the $L^2$-Fourier regression, setting $\lambda = \Theta(n^{-2s/(2s+d)})$ and taking $m = n^{1/(2s+d)}$ leads to Sobolev minimax rate.

### C.3 PROOF OF PROPOSITION 5.1 (AND ITS EXTENDED VERSION PROPOSITION A.2)

We rely on the same tools as in the proof of Propositions 3.1 and 3.2. The key is to remark that the empirical risk

$$R(\theta) = \|\Phi_1 \theta_1 + \cdots + \Phi_d \theta_d - \mathbb{Y}\|_2^2 + \lambda \|\theta\|_2^2$$

can be written as

$$R(\theta) = \|\Phi\theta - \mathbb{Y}\|_2^2 + \lambda \|\theta\|_2^2$$

where $\Phi = (\Phi_1 \mid \cdots \mid \Phi_d)$ and $\theta = (\theta_1, \ldots, \theta_d) \in \mathbb{C}^{d(2m+1)}$. By setting $\varphi(x) = (\phi(x), \ldots, \phi(x)) \in \mathbb{C}^{d(2m+1)}$, we have that $f_\theta(x) = \langle \varphi(x), \theta \rangle$. The empirical covariance matrix is then $\hat{\Sigma} = \Phi^\star \Phi / n = n^{-1} \sum_{j=1}^n \varphi(X_j) \varphi(X_j)^*$.

**Risk decomposition.** As in the proof of Propositions 3.1 and 3.2, we have, for any $\theta \in \mathbb{C}^{d(2m+1)}$,

$$\mathbb{E}(\|f_{\hat{\theta}} - f^\star\|_{L^2(\mathbb{P}_X)}^2) \le 2\mathbb{E}(\|f_{\hat{\theta}} - f_\theta\|_{L^2(\mathbb{P}_X)}^2) + 2\mathbb{E}(\|f_\theta - f^\star\|_{L^2(\mathbb{P}_X)}^2),$$

and

$$\mathbb{E}(\|f_{\hat{\theta}} - f_\theta\|_{L^2(\mathbb{P}_X)}^2) \le 3\mathbb{E}(\|\sqrt{\Sigma}(\hat{\Sigma} + \lambda)^{-1}\lambda\theta\|_2^2) + 3\mathbb{E}(\|\sqrt{\Sigma}(\hat{\Sigma} + \lambda)^{-1}\Phi^*\epsilon/n\|_2^2)$$
$$+ 3\mathbb{E}(\|\sqrt{\Sigma}(\hat{\Sigma} + \lambda)^{-1}\Phi^*\delta\mathbb{Y}/n\|_2^2),$$

where $\Sigma = \mathbb{E}(\phi(X)\phi(X)^*)$. Similarly, the bias term is bounded by

$$\mathbb{E}(\|\sqrt{\Sigma}(\hat{\Sigma} + \lambda)^{-1}\lambda\theta\|_2^2) \le \lambda(1 + \frac{\alpha^2}{\lambda n})^2 \|\theta\|_2^2,$$

where $\alpha = \max_{x \in \mathbb{R}^d} \|\phi(x)\|_2 = \sqrt{d(2m+1)}$. As for the variance term, it is bounded by

$$\mathbb{E}(\|\sqrt{\Sigma}(\hat{\Sigma} + \lambda)^{-1}\Phi^*\varepsilon/n\|_2^2) \le \sigma^2 n^{-1}(1 + \frac{\alpha^2}{\lambda n}) \operatorname{tr}(\mathbb{E}((\Sigma + \lambda)^{-1}\Sigma)) \le \sigma^2 \frac{d(2m+1)}{n}(1 + \frac{\alpha^2}{\lambda n}).$$

Finally, the approximation term is bounded by

$$\mathbb{E}(\|\sqrt{\Sigma}(\hat{\Sigma} + \lambda R)^{-1}\Phi^*\delta\mathbb{Y}/n\|_2^2) \le (1 + \frac{\alpha^2}{\lambda n})^2 \, \mathbb{E}(\|f^\star - f_\theta\|_{L^2(\mathbb{P}_X)}^2).$$

Putting everything together, we obtain

$$\mathbb{E}(\|f_{\hat{\theta}} - f^\star\|_{L^2(\mathbb{P}_X)}^2) \le \inf_{\theta \in \mathbb{C}^{d(2m+1)}} \left( (2 + 6(1 + \frac{\alpha^2}{\lambda n})^2) \mathbb{E}(\|f^\star - f_\theta\|_{L^2(\mathbb{P}_X)}^2) + \lambda(1 + \frac{\alpha^2}{\lambda n})^2 \|\theta\|_2^2 \right)$$
$$+ \sigma^2 \frac{\alpha^2}{n}(1 + \frac{\alpha^2}{\lambda n}).$$

**Convergence rate.**    The convexity of the function $x \mapsto x^2$ leads to

$$\mathbb{E}(\|f^\star - f_\theta\|_{L^2(\mathbb{P}_X)}^2) = \mathbb{E}(\|\sum_{\ell=1}^{d} g_\ell^\star - g_{\theta_\ell}\|_{L^2(\mathbb{P}_X)}^2) \leq d \sum_{\ell=1}^{d} \mathbb{E}(\|g_\ell^\star - g_{\theta_\ell}\|_{L^2(\mathbb{P}_X)}^2),$$

where $g_{\theta_\ell}(x) = \langle \phi_\ell(x), \theta_\ell \rangle$. According to Proposition A.1, taking $m = n^{1/(2s+1)}$ ensures that

$$\inf_{\theta_\ell \in \mathbb{C}^{2m+1}} \left( \mathbb{E}(\|g_\ell^\star - g_{\theta_\ell}\|_{L^2(\mathbb{P}_X)}^2) + n^{-2s/(2s+1)}\|\theta_\ell\|_2^2 \right) = \mathcal{O}(n^{-2s/(2s+1)}).$$

Thus, choosing $\lambda = n^{-2s/(2s+1)}$ and $m = n^{1/(2s+1)}$, we are led to

$$\inf_{\theta \in \mathbb{C}^{d(2m+1)}} \left( (2 + 6(1 + \frac{\alpha^2}{\lambda n})^2)\mathbb{E}(\|f^\star - f_\theta\|_{L^2(\mathbb{P}_X)}^2) + \lambda(1 + \frac{\alpha^2}{\lambda n})^2\|\theta\|_2^2 \right) = \mathcal{O}(n^{-2s/(2s+1)}).$$

We conclude that

$$\mathbb{E}(\|f_{\hat\theta} - f^\star\|_{L^2(\mathbb{P}_X)}^2) = \mathcal{O}(n^{-2s/(2s+1)}).$$

