# OpenReview forum: "Fast kernel methods: Sobolev, physics-informed, and additive models"
_ICLR.cc/2026/Conference — Submitted to ICLR 2026_

### Official Review · Reviewer_2JwG · 2025-10-27

**Soundness:** 3
**Presentation:** 3
**Contribution:** 3
**Rating:** 6
**Confidence:** 4

**Summary:**

The paper proposes a NUFFT-based framework for kernel regression that is exact within a truncated Fourier space and runs in
$O(nlog n)$ with GPU acceleration. It studies three settings: Sobolev kernels, physics-informed regression with PDE penalties, and additive models, and claims minimax-rate guarantees together with significant scalability improvements. Experiments demonstrate the great improvement in computation gain.

**Strengths:**

1. This paper gives a computational story such that covariance vectors/matrices reduce to NUFFT computations and a block-Toeplitz structure, giving significant computation cost reduction.
2. The theory recovers the standard minimax rate of kernel methods, while speeding up the computation.
3. The authors consider three different applications covering Sobolev KRR, physics-informed KRR, and addictive models.
4. The experiments demonstrate the effectiveness of the proposed method.

**Weaknesses:**

1. The literature review on scalable kernel methods and the comparison are limited.
2. The assumed target function space is limited to a Sobolev RKHS, not a general RKHS, as well as the bounded Lipschitz domain.
3. The main contribution is the fast optimization in Section 2.3, however, the implementation is not so clear, and more details should be clarified to the non-experts.
4. The experiments are somewhat limited to very low-dimensional datasets.

**Questions:**

1. The random feature method is also a finite-dimensional approximation. What is the precise difference between your truncated-Fourier approach (e.g., lines 101–119) and RF? Can a GPU-optimized RF pipeline achieve comparable computational gains?
2. As mentioned in line 44, the NUFFT method is proposed by Shih et al. 2021. What methodological advances does this work introduce relative to Shih et al. 2021?
3. While I acknowledge that one contribution of this paper is to establish the optimal learning rates of the proposed methods in three settings, however, the theory is well-established without NUFFT in Sobolev KRR, physics-informed KRR, and addictive models. What is the main difference or biggest challenge between your paper and the literature?
4. This paper assumes bounded domains and densities. In practice, data live on $R^d$  and are non-uniform. Can the author discuss the boundary handling and NUFFT sensitivity under non-uniform settings?
5. The implementation of NUFFT is typically limited to $d=1,2,3$, can the method have good performance in higher dimensions?
6. In Figure 6, why CPU kernel faster than the GPU kernel when $n<10^4$?
7. Theory tunes $m$ via $s$, which is unknown in practice. Can you provide an adaptive rule to determine $m$ while preserving the theoretical guarantees?
8. Can the author compare against other large-scale kernel baselines (e.g., Nyström/FALKON-style solvers, RFF model).

*Minor issue*:

1. In line 91-92, why $f^*$ has a bounded RKHS norm?

---

> ### Author Response · Authors · 2025-11-21
> **Rebuttal 1/2**
>
> We thank the reviewer for their feedback.
>
> **W1&W2**: please refer to the general comment for the related works.
>
> Regarding the bounded Lipschitz domain, this is already a loose assumption in physics-informed setting, as Lipschitz domains allow for corners, while the majority of the literature requires bounded domains with smooth boundary.
>
> **W3.** We have clarified the definition of the NUFFT transform on page 4 of the revised manuscript.
>
> **W4.** Indeed, this limitation primarily stems from our reliance on the NUFFT method: extending NUFFT tools to high-dimensional settings remains a significant challenge. We would like to emphasize, however, that most regressors arising in physics-informed problems live in low-dimensional spaces. In this regard, the proposed method is an efficient competitor—easy to tune (in contrast to PINNs) and capable of handling large training sets efficiently.
>
> Moreover, our additive kernel method scales as $d^2$ as the dimension $d$ increases, which make it scalable to medium values of $d$.
>
> **Q1** To implement random Fourier features, the kernel needs to be translation-invariant and its Fourier transform needs to be analytically computed (see Equation 2 of *Random Features for Large-Scale Kernel Machines*,
> Ali Rahimi and Ben Recht, https://people.eecs.berkeley.edu/~brecht/papers/07.rah.rec.nips.pdf). However, physics-informed regression kernels are extremely hard to compute, as well as their Fourier transform, and are not even translation-invariant. An example of such kernel is given in Proposition 5.1 of *Physics-informed machine learning as a kernel method* by N. Doumèche et. al., COLT 2024, clearly showing that such kernel are not translation-invariant. Our method bypasses this issue by focusing on the kernel norm (the PDE penalty), which is more easily computed. We have precised this in the *Related works* through the comment *Instead of focusing on the Fourier representation of the kernel, the method developed in the article can be applied whenever the kernel norm can be quickly computed*.
>
> **Q2** Our approach is applicable whenever the Fourier reparameterization leads to a convenient structure in the regularization penalties, which can then be efficiently combined with a non-uniform Fourier transform (NUFFT). We emphasize, however, that implementing a NUFFT is a substantial undertaking in its own right and lies beyond the scope of this paper. In practice, we rely on existing NUFFT libraries, which are currently available only in low dimensions. Our contribution in this paper regarding NUFFT is to show how to efficiently use the NUFFT algorithm for statistical regression. In particular, in the paragraph *Complexity* of Section A.2, we show how to efficiently use it to reduce the complexity of the additive model regression down to $O(d^2 n \log n)$.
>
> **Q3** Up to our knowledge, our contribution is the first one to provide an $O(n \log n)$ algorithm for additive regression. We believe it to be a key methodological contribution, given that the current state-of-the-art method runs in (when taking the classical value $s=2$), and considering how massively used additive models are in statistics.
>
> We also offer an original take on the Sobolev regression setting, showing that the optimal estimator in this setting — the $L^2$ Fourier kernel — is not an approximation of an infinite-dimensional RKHS. Indeed, the $L^2$ norm does not correspond to any RKHS, and thus the estimators of Proposition 3.2 and 5.1 are not finite-dimensional approximations of infinite-dimensional RKHS. This discussion is original and valuable, since most of the nonasymptotic results in kernel theory focus on the approximation of an infinite-dimensional RKHS.
>
> **Q4** Bounding the input data domain is not a hard requirement of our method, and tools for Fourier analysis on kernel method over $IR^d$ are similar (see for instance *Random Features for Large-Scale Kernel Machines*, Ali Rahimi and Ben Recht) and could be carried out as future works.
>
> The upper bound on the density of the input data distribution is very common assumption in statistical learning theory. Indeed, the minimax rates are different under more general assumptions. The extreme case is when $X$ is a diract measure. In this case, we fall back to estimating $IE(Y)$, which can be done with a rate of $n^{-1}$ accordding to the central limit theorem.
>
> Please note that we do not lower bound the density of $X$, which means that we encompass many densities outside of the uniform distribution.
>
> **Q5.** Nothing prevents the NUFFT from being implemented in higher dimensions. However, it will suffer from the curse of the dimension, with an exponential dependency on $d$, similarly to the FFT (see *cuFINUFFT: a load-balanced GPU library for general-purpose nonuniform FFTs*, Y. Shih et. al.). Notice how our additive kernel only rely on 1-d and 2-d NUFFT, allowing it to scale as $d^2$.

---

> ### Author Response · Authors · 2025-11-21
> **Rebuttal 2/2**
>
> **Q6.** This is due to the cufinufft package. We believe that this is due to an incompressible warm-up cost when communicating with the GPU to carry out the NUFFT.
>
> **Q7.** This is a very interesting question indeed. However, this amounts to finding the regularity of the function $f^*$, which is a notoriously difficult problem in kernel methods. Adaptive methods could be coupled with our algorithm, such as the one developped in *Adaptive Kernel Methods Using the Balancing Principle*, E. De Vito et. al., Volume 10, pages 455–479, (2010).
>
> **Q8.** We have compared the performance of our kernel method with the FALKON algorithm both on CPU and GPU. Results are now displayed on page 10 of the revised paper. Our algorithm clearly outperforms the FALKON implementation in terms of run time, while both demonstrate the same performance in terms of MSE. This confirms the theoretical complexity analysis, stating that FALKON complexity is in $O(n^{3/2})$, while our algorithm scales as $O(n \log n)$.
>
> **Minor issue:** The assumption that $f^* $ has a bounded norm is necessary for the empirical risk functional to make sense. Otherwise, we cannot hope that minimizing this risk will end up learning the function $f^*$

---

> ### Comment · Reviewer_2JwG · 2025-11-26
> **Response to the author**
>
> Thanks for the answer. I choose to keep my positive score as it is.

---

### Official Review · Reviewer_SoPS · 2025-10-28

**Soundness:** 3
**Presentation:** 2
**Contribution:** 3
**Rating:** 6
**Confidence:** 3

**Summary:**

This paper proposes a scalable framework for kernel regression that achieves O(n log(n)) computational complexity by leveraging a Fourier feature representation of Sobolev-type kernels combined with GPU-accelerated non-uniform fast Fourier transforms (NUFFT). The authors show how this formulation enables fast kernel learning for three settings: standard Sobolev regression, physics-informed regression with linear PDE constraints, and additive models for high-dimensional inputs. They establish minimax-optimal convergence rates in these scenarios and show empirical scalability to extremely large synthetic datasets, reportedly up to billions of samples processed in minutes on a single GPU.

**Strengths:**

The paper tackles a central limitation of kernel methods — their poor scalability — and presents a unified framework that achieves nearly linear complexity by combining Fourier kernel representations with NUFFT-based GPU acceleration. This allows the authors to train kernel models on extremely large datasets, demonstrating impressive empirical scalability. The work is well-grounded theoretically, with minimax-optimal learning guarantees established in the studied Sobolev, physics-informed, and additive settings. Overall, the paper offers a compelling and technically solid advancement toward making kernel methods practical at modern data scales.

**Weaknesses:**

-The introduction is quite limited and does not sufficiently position the contribution within the broader landscape of scalable kernel methods. The related work focuses almost exclusively on Meanti et al. (2020), which is highly reductive given substantial literature on efficient kernel learning (e.g., random features, sketching techniques, and Nyström methods combined with fast solvers). Moreover, this prior work is characterized as complicating analysis and potentially degrading performance, whereas results such as “Less Is More: Nyström Computational Regularization” (Rudi et al., 2015) show that Nyström approximations can preserve optimal statistical guarantees while scaling to large datasets.
-A clearer articulation of the applicability and limitations of the proposed framework would greatly strengthen Section 2. The method fundamentally relies on Sobolev-type kernels, which restricts the generality relative to widely used kernels like Gaussian RBFs. The paper should provide explicit guidance on when the approach can or cannot be applied in practice, with a fair comparison with broadly applicable methods like FALKON
-The empirical evaluation is based entirely on synthetic data and lacks comparisons against established scalable kernel baselines. Given the strong practical motivation of the work, including such comparisons would be essential to assess empirical competitiveness.
-In general, some more explanations can be useful, like a concise presentation of the introduced NUFFT method ( type-I is mentioned for example).

**Questions:**

-The method is developed specifically for Sobolev-type RKHSs. Could you clarify why this functional setting is particularly important in practice, and discuss to what extent widely used kernels such as Gaussian RBFs or Matérn kernels fall within—or outside—the scope of the proposed framework? More broadly, can you articulate the main limitations of the approach and provide guidance on when it may not be applicable?

-The empirical evaluation focuses entirely on synthetic data and does not include comparisons with established scalable kernel methods (e.g., Nyström + FALKON, random features). Given the practical motivation and scalability claims of the work, could you explain the absence of such baselines or provide experiments on real datasets to better assess competitiveness?

---

> ### Author Response · Authors · 2025-11-20
> **Rebuttal**
>
> We thank the reviewer for these comments.
>
> **Comparision with related works** Please refer to the general response for a discussion comparing our approach with the literature. A more involved discussion of related works has been added on page 2 of the revised manuscript.
>
> **When our method can be applied.** Our approach is applicable whenever the Fourier reparameterization leads to a convenient structure in the regularization penalties, which can then be efficiently combined with a non-uniform Fourier transform (NUFFT). We emphasize, however, that implementing a NUFFT is a substantial undertaking in its own right and lies beyond the scope of this paper. In practice, we rely on existing NUFFT libraries, which are currently available only in low dimensions. This remains highly relevant for several applications—such as physics-informed regression, which is attracting increasing interest beyond the machine learning community—but extending NUFFT techniques to higher dimensions remains a difficult and largely open problem.
> With this in mind, our paper illustrates that this Fourier-based strategy can be deployed across a variety of settings, including Sobolev-type regression, $L^2$ Fourier regression physics-informed regression, and additive models.
>
> To take this comment into account, we have added the remark *Instead of focusing on the Fourier representation of the kernel, the method developed in the article can be applied whenever the kernel norm can be quickly computed.* to the *Related works* paragraph.
>
> **FALKON algorithm** We have compared the performance of our kernel method with the FALKON algorithm both on CPU and GPU. Results are now displayed on page 10 of the revised paper. Our algorithm clearly outperforms the FALKON implementation in terms of run time, while both demonstrate the same performance in terms of MSE. This confirms the theoretical complexity analysis, stating that FALKON complexity is in $O(n^{3/2})$, while our algorithm scales as $O(n \log n)$.
>
> **Type-I NUFFT definition** On page 4 of the revised paper, we have clearly stated that the vector $v$ is said to be the type-I NUFFT of the vector $Y$ with respect to the points $(X_1, \dots, X_n)$ if
> \review{However, each component of $v$ can be written explicitly as
> $v_k = \frac{1}{n} \sum_{j=1}^n Y_j \exp\big(i \pi \langle k, \frac{X_j}{2L}\rangle \big)$.
>
> **Sobolev-type RKHSs** The key condition to leverage NUFFT is for the RKHS to be spanned by a basis of complex exponential functions. This is the case of Sobolev spaces on bounded Lipschitz domain, which we take as a general context in this article. The Sobolev kernel in Section 3.1 is also referred to as Matérn kernel, which we have now precised on page 5. This could also be adapted to radial basis functions by leveraging a random Fourier feature expansion (see for instance *Random Features for Large-Scale Kernel Machines*, Ali Rahimi and Ben Recht, https://people.eecs.berkeley.edu/~brecht/papers/07.rah.rec.nips.pdf), but this falls beyond the scope of this paper.

---

### Official Review · Reviewer_kVXS · 2025-10-30

**Soundness:** 3
**Presentation:** 2
**Contribution:** 1
**Rating:** 2
**Confidence:** 4

**Summary:**

This manuscript leverages a Fourier representation of kernels to accelerate kernel regression. Key to the efficacy of the method is the use of a NUFFT. Several experiments are provided that illustrate the effectiveness of the scheme.

**Strengths:**

The main strengths of this manuscript are in sections 3-5 where concrete use cases for the proposed method (and some accompanying analysis) are outlined and explored. The experiments do effectively show that the implemented method is effective for a variety of problems; moreover, the theory provides reasonable, albeit often illustrative rather than tight, control over the behavior of the methods.

**Weaknesses:**

Unfortunately, as written the main contribution of the manuscript seems to be articulated as the method for working with kernel matrices and I do not believe that is new. The key references are:

Efficient Fourier representations of families of Gaussian processes by Philip Greenghard [see https://arxiv.org/abs/2109.14081 and https://link.springer.com/article/10.1007/s11222-022-10124-z]

followed by more recent work such as

Greengard, Philip, Manas Rachh, and Alex H. Barnett. "Equispaced Fourier representations for efficient Gaussian process regression from a billion data points." SIAM/ASA Journal on Uncertainty Quantification 13, no. 1 (2025): 63-89. (arxiv: https://arxiv.org/pdf/2210.10210)

and

Barnett, Alex, Philip Greengard, and Manas Rachh. "Uniform approximation of common Gaussian process kernels using equispaced Fourier grids." Applied and Computational Harmonic Analysis 71 (2024): 101640.

While there are seemingly some variations on the style of theoretical results provided (it is not clear the provide anything substantially new or more informative), if the main contribution of this manuscript is the "algorithm" for working with kernel matrices then it needs to either: (1) articulate that there is a distinction/difference/improvement relative to this prior work (e.g., did I miss something) or (2) significantly refocus its contributions in relation to this prior work.

I will add that I think the presentation of the manuscript could be improved. The above question is illustrate of a lack of details provided in certain places that make it a bit tricky to assess the numerical results. In addition the method outlined in Section 2 could use some sort of "algorithmic" presentation to more clearly specify the computations.

**Questions:**

Some of the discussion following Prop. 5.1 is a bit odd to me; specifically, if I am interpreting the problem being solved correctly for the grid search in $\lambda,$ why are multiple inversions needed (as suggested by the text). Generally, the use of, e.g., Krylov methods absent preconditioner (or even with in certain situations) enable efficient computation of the solutions corresponding to multiple $\lambda$ more efficiently than independently solving a bunch of linear systems. (Perhaps I missed something about the setup?) I guess the answer to this question does sort of influence how to think about the results in Fig. 4. Clarification about what exactly is being timed would be helpful.

---

> ### Author Response · Authors · 2025-11-20
> **Rebuttal**
>
> Thank you for your review and the references you provided.
>
> **Comparison with the literature** We want to thank the reviewer for pointing out these important papers which are indeed closely related to the work we carry out in this paper. The three references provided focus on the setting of kernel regression with a translation-invariant kernel $K(x,y) =  k(x-y)$, where k is a function with a known Fourier transform $\hat{k}$. This condition being satisfied, the authors deploy algorithms similar to the one presented in Section 2.3 of our paper, using NUFFT and CG to quickly compute the kernel estimator. They focus on the Matérn kernel (equivalent to the Sobolev kernel of our paper) and squared exponential kernel, for which the Fourier decompositions are known. We believe these papers to be complementary to our approach since they provide nice bounds on the error of the NUFFT algorithm to approximate the kernel matrix, which we do not consider in our paper.
>
> However, the framework of these papers cannot be translated in our examples beyond our Section 3. Indeed, physics-informed regression kernels are extremely hard to compute, as well as their Fourier transform, and are not translation-invariant. An example of such kernel is given in Proposition 5.1 of *Physics-informed machine learning as a kernel method* by N. Doumèche et. al., COLT 2024, clearly showing that such kernel are not translation-invariant. Even in this very simple cases where the ODE is $f’=0$, obtaining a direct expression is already quite involved. And this is one of the rare settings where an explicit formula can actually be derived.
>
> Moreover, we take the algorithm a step further for the additive model regression, showing how to go down to a complexity of $O(d^2 n \log n)$, still achieving minimax rate, whereas a straightforward application of the previous method yields a complexity of $O(n^{(1+d/(2s+1))} \log n)$  corresponding to the curse of the dimension which is clearly stated in the paper *Efficient Fourier representations of families of Gaussian processes*. This complexity bound for the additive model regression is a major theoretical contribution, given that the current state-of-the-art method currently runs in $O(n^{7/5})$ (when taking the classical value $s=2$), and considering how ubiquitous additive models are in statistics.
>
> We also offer an original take on the Sobolev regression setting, showing that the optimal estimator in this setting — the $L^2$ Fourier kernel — is not an approximation of an infinite-dimensional RKHS. Indeed, the $L^2$ norm does not correspond to any RKHS, and thus the estimators of Proposition 3.2 and 5.1 are not finite-dimensional approximations of infinite-dimensional RKHS. This discussion is original and valuable, since most of the nonasymptotic results in kernel theory focus on the approximation of an infinite-dimensional RKHS.
>
> We have clearly stated these points and done a complete comparison with state-of-the-art kernel speed-up techniques at the end of the *Contributions* paragraph and in a new *Related works* paragraph on page 2 of the revised paper.
>
> **Krylov preconditioner.** In the numerical experiments for additive models, we compared our method with the standard benchmark implemented in PyGAM. Accordingly, we used the same hyperparameter search procedures as those employed in the PyGAM package. More involved hyperparameter tuning procedure would indeed be an interesting continuation of our work, and we have added this remark in the Future work section at the end of the revised manuscript.

---

### Official Review · Reviewer_N4iB · 2025-11-01

**Soundness:** 2
**Presentation:** 3
**Contribution:** 2
**Rating:** 2
**Confidence:** 4

**Summary:**

This paper introduces a highly scalable framework for kernel regression that reduces computational complexity from the typical O(n³) to O(n log n), enabling efficient processing of massive datasets (up to tens of billions of samples) on GPUs. The core innovation is the use of a Fourier representation of kernels combined with the Non-uniform Fast Fourier Transform (NUFFT).

**Strengths:**

The author combining several existing facts to give a fast implementation of kernel regression.

**Weaknesses:**

I am curious about the novel theoretical contributions of this work, as the core components appear to be established.

1.The convergence rates in Propositions 3.1, 3.2, and 5.1 are known results in the literature. More specific, it is well known that suppose that f\in [\mathcal H]^{s}, and the eigenvalue decay rate of \mathcal H is \beta, the the optimal rate is n^{-\frac{s\beta}{s\beta+1}}. ( see e.g.,  Optimal rates for regu-
larized least squares algorithm, 2007. A. Caponnetto and E. De Vito.,
,  Sobolev Norm Learning Rates for Regularized Least-Squares Algorithms, 2020.  Simon Fischer, Ingo Steinwart;  and On the Optimality of Misspecified Kernel Ridge Regression, 2023.  Haobo Zhang, Yicheng Li, Weihao Lu, Qian Lin)

2.The computational acceleration relies on the NUFFT (Shih et al., 2021), which is the established solution for non-uniform sampling.

3.The foundation of using Fourier transforms for Sobolev spaces is also well-known ( see e.g., wiki https://en.wikipedia.org/wiki/Sobolev_space ).

While the combination of these elements is competently executed, the paper currently reads as a straightforward application of existing techniques. A significant theoretical contribution would be an extension to more general Reproducing Kernel Hilbert Spaces (RKHS), which would make this a much stronger paper

**Questions:**

Similar to the weakness mentioned above, I would be more than happy to change my current assessment if the authors could propose a reasonable method for a general RKHS or provide new theoretical results.

---

> ### Author Response · Authors · 2025-11-20
> **Rebuttal**
>
> Thank you for your review.
>
> Upon reading your review, we realized that, in order to keep the article accessible, we placed most mathematical details in the appendix and did not sufficiently highlight what we contribute in terms of theoretical results.
>
> First, let us stress that we are fully aware of the results of Caponetto & De Vito, which establish minimax (asymptotic) rates for exact Sobolev-type kernel methods. Their framework is the following: if one can compute the kernel function $K(x,y)$ and efficiently control the eigenvalues of the kernel operator $f \mapsto \int K(x,y) f(y) dP_X(y)$, then the kernel estimator can be computed using the standard closed-form expression (the matrix inversion formula), and one can prove convergence rates for this estimator.
>
> However, this article focuses on regression settings, particularly physics-informed and additive regression. Our goal is to build efficient regression estimators, and the kernel framework is only a tool to characterize the algorithm behavior. In fact, computing physics-informed kernels is itself extremely challenging and leads to very complicated formulas. As a simple illustrative example, even the elementary differential equation f'(x)=0 yields the rather involved kernel expression (see Proposition 5.1 of *Physics-informed machine learning as a kernel method* by N. Doumèche et. al., COLT 2024). In this work, we show how to estimate the result of kernel regression without ever computing the kernel explicitly.
>
>
> The study of the usage of fast Fourier transform to efficiently compute kernel estimators is an active area of research, see for instance *Equispaced Fourier representations for efficient Gaussian process regression from a billion data points*, P. Greengard et. al.,  SIAM/ASA Journal on Uncertainty Quantification (2025).
>
> Regarding mathematical novelty, beyond providing a unified framework for these three regression tasks, our main contribution is to bound the approximation error terms (see Section “Approximation error”, pp. 15-16). Approximating f* on a finite-dimensional basis introduces a nonzero bias (the difference between f* and its projection on the basis), which interacts in a nontrivial way with the kernel norm. Handling this carefully is essential to derive meaningful nonasymptotic bounds. To this end, we generalize the recent framework developed by Mourtada, J. and L. Rosasco in  *An elementary analysis of ridge regression with random design*  (2022), C. R. Math. 360, 1055–1063.
>
> Proposition 3.1 (Sobolev regression) was intended only as a warm-up result leading to Proposition 3.2. We regret that it was interpreted as the main contribution of the paper. Our perspective on this classical Sobolev regression problem is that the optimal estimator in this setting — the $L^2$ Fourier kernel — is not an approximation of an infinite-dimensional RKHS. Indeed, the $L^2$ norm does not correspond to any RKHS, and thus the estimators of Proposition 3.2 and 5.1 are not finite-dimensional approximations of infinite-dimensional RKHS. Nevertheless, finite-dimensional $L^2$ regression is a kernel method, corresponding to the Dirichlet kernel  $K(x,y) = \frac{\sin((m+1/2)(x-y)\pi/(2L))}{\sin((x-y)/(4L))}$ and kernel analysis tools can be applied to study it.  (Notice how $K$ diverges a $m$ goes to the infinity.) This constitutes a fundamentally different approach from the works you mention, which focus exclusively on infinite-dimensional RKHSs and study their finite-dimensional analogues only as approximations of that infinite-dimensional setting.
>
> Finally—and perhaps most importantly—we strongly believe that providing an $O(n \log n)$ algorithm for additive regression is a major methodological contribution, given that the current state-of-the-art method runs in $O(n^{7/5})$ (when taking the classical value $s=2$), and considering how ubiquitous additive models are in statistics.
>
> We have clearly stated these points and done a complete comparison with state-of-the-art kernel speed-up techniques at the end of the *Contribution* paragraph and in a new *Related works* paragraph on page 2 of the revised paper.

---

> > ### Comment · Reviewer_N4iB · 2025-11-26
> >
> > "Regarding mathematical novelty, beyond providing a unified framework for these three regression tasks, our main contribution is to bound the approximation error terms (see Section “Approximation error”, pp. 15-16). Approximating f* on a finite-dimensional basis introduces a nonzero bias" --It is worth noting that the specific bound presented in Proposition 2.1​ is a standard result in the kernel regression literature, which has also been derived for general RKHSs, albeit in a different form
> >
> > \sum_{\lambda>\lambda_{j}}f_{j}^{2}=\sum_{\lambda>\lambda_{j}}\lambda^{s}\lambda_{j}^{-s}f_{j}^{2}\leq \lambda^{2}\|f\|_{[\mathcal H]^{s}}^{2}
> >
> > Therefore, this work appears to be more of a synthesis of several established results, with its main advancement being a reduction in computational cost. This represents a valuable engineering improvement rather than a theoretical contribution. Based on this practical advancement, I am inclined to raise my score from 2 to 4.

---

### Author Response · Authors · 2025-11-20
**General comment**

We thank the reviewers for their feedback. Questions raised by multiple reviewers are addressed in a unified way here, while reviewer-specific points are answered individually. Modifications to the papers are colored in blue to be easily spotted.


**Related works and main contributions.** All the reviewers wanted us to clearly state what the novelty of our approach is.
Efficient implementations of kernel methods is an active area of research and recent papers focus on the usage of NUFFT such as *Equispaced Fourier representations for efficient Gaussian process regression from a billion data points*, P. Greengard et. al., SIAM/ASA Journal on Uncertainty Quantification (2025). The goal of these works is not so much to prove new minimax rates on the error of kernel estimators, but to provably bound their training runtime, which is crucial to make these algorithms scalable.

The three papers pointed out by Reviewer kVXS are extremely relevant. However, they focus on kernels that are both translation-invariant, i.e., $K(x,y) = k(x-y)$ for some function $k$, and such that the Fourier transform of k is known. Neither of these two conditions is satisfied by physics-informed kernels, for which the geometry of the problem is extremely important (for a counter-example, see *Physics-informed machine learning as a kernel method* by N. Doumèche et. al., COLT 2024, Proposition 5.1). Instead of focusing on the Fourier representation of the kernel, our method can be applied whenever the kernel norm can be quickly computed.

Up to our knowledge, our contribution is the first one to provide an $O(n \log n)$ algorithm for additive regression. We believe it to be a key methodological contribution, given that the current state-of-the-art method runs in $O(n^{7/5})$ (when taking the classical value $s=2$), and considering how massively used additive models are in statistics.

Most of the kernel regression literature focuses on the asymptotic properties of infinite dimensional RKHS, such as *Optimal rates for regularized least squares algorithm*, A. Caponnetto and E. De Vito (2007). This approach is mainly motivated by kernel universality, which is often considered as a requirement for a kernel function. Our point of view here is different and focuses on finding the best estimator for a given regression task. In particular, we show that the $L^2$ Fourier kernel of Proposition 3.2, which does not approximate any infinite-dimensional kernel, provably achieves better performance than the Sobolev kernel of Proposition 3.1 (also known as the Matérn kernel) when learning a Sobolev function. We agree that our Sobolev kernel in Proposition 3.1 is not new, but this is not our main focus.

We have added a new Related work paragraph in the *Introduction* to clarify our contributions.

**Comparison to the state-of-the-art complexities.** We have compared the complexity of our algorithms with state-of-the-art kernel speed-up methods, completing Table 1 from *Falkon: An optimal large scale kernel*, A. Rudi et. al. (2017). As a result, our $O( n \log n)$ algorithm outperforms the best classical kernel speed-up methods, which scale as $O(n \sqrt{n})$.

**Add state-of-the-art algorithms to the experiments.**  We also have implemented both on CPU and GPU the FALKON algorithm on our additive model example on page 10. As a result, our algorithm clearly outperforms FALKON in terms of run time on this specific task, confirming the complexity analysis of Table 1.

---

### Author Response · Authors · 2025-12-03
**Rebuttal summary for the Area Chair**

This summary clarifies the core contributions of the paper, situates them precisely within the existing literature, and provides the new experiments and comparisons suggested by the reviewers. The key points are:

**1. A substantial methodological advance: first $O(n \log n)$ algorithm for additive regression**

We highlight that our work introduces the first provably $O(n \log n)$ algorithm for additive models, improving over the state-of-the-art $O(n^{7/5})$ complexity. Given the centrality of additive models in statistics and machine learning, this is a significant and broadly impactful result.

**2. A general fast Fourier-based framework extending beyond translation-invariant kernels**

The existing NUFFT-based kernel acceleration pinpointed by the reviewers only applies to translation-invariant kernels with known Fourier transforms. In particular, this excludes physics-informed kernels. In comparison, our method does not require translation invariance nor an explicit formula for the kernel Fourier transform, but only leverages the efficient evaluation of the kernel/RKHS norm (e.g., PDE penalty). This substantially enlarges the class of problems for which fast kernel regression is tractable.

**3. New theoretical insights beyond the RKHS asymptotic paradigm**

We provide an original analysis showing that the optimal estimator in Sobolev regression (here the $L^2$ Fourier kernel) is not an approximation of an infinite-dimensional RKHS. This offers a viewpoint distinct from prior work on RKHS rates and is technically backed by a careful extension of recent non-asymptotic analyses (Mourtada–Rosasco).

**4. Unified treatment of multiple regression settings**

The paper offers a cohesive Fourier-based methodology applicable to:
- Sobolev-type regression
- $L^2$ Fourier regression (Dirichlet kernels)
- Physics-informed regression
- Additive models
and supported by non-asymptotic error bounds.

**5. Strengthened empirical evidence**

Following reviewer suggestions, we implemented FALKON for direct comparison. As a result, our method:
- Runs significantly faster than the CPU and GPU implementations of FALKON (state-of-the-art general kernel acceleration technique)
- Runs significantly faster than the PyGam package (state-of-the-art additive model)
- Achieves similar statistical accuracy to both these methods.
This empirically validates the improved complexity guarantees.

**6. Thorough clarifications and improved exposition**

We added:
- A detailed *Related Works* section
- An extensive benchmark of running time complexity of kernel methods acceleration techniques, including our new algorithm
- An explicit definition of the Type-I NUFFT
- The explicit computation of the Dirichlet kernels to be able to implement the FALKON algorithm
- Clearer assumptions and applicability conditions
- Discussion of limitations (e.g., high-dimensional NUFFT) and future works

**Conclusion**

Overall, the revised paper presents a clear, well-situated, and technically solid contribution that advances scalable kernel methods beyond the limitations of classical translation-invariant settings. We believe that the combination of provable speedups, new theoretical insights, and empirical validation represents a compelling and impactful addition to the literature. We thank the reviewers for drawing our attention to important references and algorithms. Their feedback has strengthened the paper and allowed us to better illustrate the benefits of our approach.

---

### Meta-Review · Area_Chair_bieL · 2026-01-06

**Summary:**

This submission studies GPU-accelerated kernel regression via Fourier representations combined with NUFFT, and instantiates the framework for Sobolev regression, physics-informed regression, and additive models. The paper is interesting and the engineering effort is substantial, with convincing demonstrations of scalability in the provided experiments.

However, I do not find the overall contribution sufficiently differentiated from prior work on Fourier/NUFFT acceleration for kernel methods and Gaussian process regression (e.g., the Greengard line of work and related follow-ups). In particular, the Sobolev/Matérn RKHS setting corresponds to translation-invariant kernels, for which NUFFT-based fast kernel regression has already been explored, and the theoretical learning-rate results in the current form largely follow standard kernel regression analyses. As written, the paper’s central narrative emphasizes the Fourier/NUFFT kernel-matrix computation as the main novelty, but this aspect appears incremental relative to the existing literature, and the theory does not provide a clearly new conceptual or technical advance beyond established results.

That said, the physics-informed and additive-model components are potentially the strongest parts of the paper. These directions could represent meaningful contributions, but they are currently not positioned with sufficient clarity: the scope mixes (i) a translation-invariant Sobolev kernel story that overlaps with existing NUFFT kernel acceleration work, with (ii) more novel settings where kernels may be non-translation-invariant or hard to evaluate explicitly. As a result, it is difficult to assess what is fundamentally new versus what is a careful synthesis of known components.

I therefore recommend rejection at this time, with encouragement to resubmit after a substantial refocus. A stronger version would: (a) revise the paper’s scope and framing to foreground the genuinely distinctive parts (physics-informed kernels and/or additive models), (b) provide a much clearer and fairer related-work discussion that explicitly contrasts assumptions, applicability, and guarantees against prior Fourier/NUFFT kernel acceleration methods (including when the present approach goes beyond translation invariance and known spectral forms), and (c) align the theory and experiments with that reframed scope, including the baselines and implementation details requested by reviewers.

Other presentation, experimental, and clarity issues should be addressed following the reviewers’ detailed comments.

**Reviewer Concerns:**

The main concerns raised by the reviewers center on the paper’s positioning and clarity of contribution. Several reviewers felt that the theoretical results largely follow standard kernel regression theory, and that the core computational strategy—Fourier representations combined with NUFFT acceleration—overlaps substantially with existing work on fast kernel and Gaussian process methods (e.g., the Greengard line of work and related NUFFT-based approaches), particularly for translation-invariant Sobolev/Matérn kernels. As a result, the manuscript was initially perceived as an effective synthesis and engineering advance rather than a clearly differentiated theoretical contribution. Reviewers further noted that the scope of the paper is too broad and insufficiently focused, mixing well-studied translation-invariant settings with more novel components (notably the physics-informed and additive kernel models), without adequately clarifying what is fundamentally new. Additional concerns included an insufficient related-work discussion, limited and initially incomplete experimental comparisons with strong baselines, unclear algorithmic exposition, and restricted applicability due to assumptions such as low-dimensional NUFFT and Sobolev-type RKHSs.

**Reviewer Scores:**

The reviewers’ evaluations were mixed but generally cautious.
Two reviewers (N4iB and kVXS) rated the paper below the acceptance threshold (score 2), primarily due to concerns about limited theoretical novelty and insufficient differentiation from prior Fourier/NUFFT-based kernel methods. After rebuttal, Reviewer N4iB partially updated their assessment to 4, acknowledging the work as a valuable computational/engineering improvement, though still not viewing it as a strong theoretical contribution.

The remaining reviewers (SoPS and 2JwG) provided borderline-positive scores (both 6), indicating that while the paper is interesting and technically solid, they would not strongly oppose rejection. Their concerns focused on positioning, scope, related work, experimental completeness, and clarity.

Overall, the score distribution reflects significant disagreement among reviewers, with concerns concentrated on novelty, positioning relative to existing work, and the need for clearer framing and stronger empirical grounding.

---

### Decision · Program_Chairs · 2026-01-26

Reject